# Estimating the optimal age for infant measles vaccination

Elizabeth Goult [1,2] ✉, Laura Andrea Barrero Guevara[1,3], Michael Briga [1,4,5,6] & Matthieu Domenech de Cellès [1]

The persistence of measles in many countries demonstrates large immunity gaps, resulting from incomplete or ineffective immunization with measles-containing vaccines (MCVs). MCV impact is determined, in part, by vaccination age. Infants who receive dose 1 (MCV1) at older ages have a reduced risk of vaccine failure, but also an increased risk of contracting infection before vaccination. Here, we designed a new method—based on a mathematical transmission model incorporating realistic vaccination delays and age variations in MCV1 effectiveness—to capture the MCV1 age risk trade-off and estimate the optimal age for recommending MCV1. We applied this method to a range of synthetic populations representing lower- and higher-income populations. We predict a large heterogeneity in the optimal MCV1 ages (range: 6–20 months), contrasting the homogeneity of observed recommendations worldwide. Furthermore, we show that the optimal age depends on the local epidemiology of measles, with a lower optimal age predicted in populations having lower vaccination coverage or suffering higher transmission. Overall, our results suggest the scope for public health authorities to tailor the recommended schedule for better measles control.

Measles is a highly contagious childhood infection[1] caused by the measles virus. The virus is primarily spread through respiratory droplets and aerosols[2], and symptoms include cough, fever, malaise, and a characteristic maculopapular rash[1]. Historically, measles was a major childhood disease, infecting almost all individuals in early life[3] with over 6 million estimated measles-related deaths occurring per year in the early 1960s[4]. The introduction of measles vaccines in the 1960s significantly reduced the global number of measles cases and deaths[5], with estimated deaths in 2021 reduced to approximately 128,000[6].

However, despite the indisputable global success of the vaccine, measles remains endemic in multiple countries. Many regional elimination targets for 2020 were not met[7], reflecting the difficulty of reaching the high immunization coverage needed for measles elimination[5]. These difficulties were compounded during the COVID-19 pandemic, which caused interruptions in routine vaccinations and

supplementary immunization activities (SIAs)[7,8], with global MCV1 coverage estimated to be 7.9% lower than expected in 2020, and particularly high disruption reported in the Global Burden of Disease super region of South Asia[8]. Between 2020 and 2022, 140 countries reported at least 1 case to the World Health Organization (WHO), and over 30 countries reported over 1000 cases in a year[9].

Although the immunity gaps that drive continued measles transmission are mainly caused by insufficient vaccine coverage, they also result from vaccine failures. Between 2013 and 2017, 25% of measles cases were attributed to measles-containing vaccine (MCV) failure after 2 doses[10]. Several causes may explain these failures (i.e, cold chain storage failure, and host-related factors such as nutrition and immune status[11]), but one key avertable cause of these vaccine failures is the vaccination age: vaccination with the first dose of measles-containing vaccine (MCV1) at younger ages results in a higher risk of vaccine

[1]Infectious Disease Epidemiology group, Max Planck Institute for Infection Biology, Charitéplatz 1, Campus Charité Mitte, Berlin, Germany. [2]Charité—Universitätsmedizin Berlin, Berlin, Germany. [3]Institute of Public Health, Charité—Universitätsmedizin Berlin, Berlin, Germany. [4]Department of Biology, University of Turku, Turku, Finland. [5]PandemiX Center of Excellence, Roskilde University, Roskilde, Denmark. [6]Charité Centre for Global Health, Charité—Universitätsmedizin Berlin, Berlin, Germany. ✉e-mail: goult@mpiib-berlin.mpg.de

failure[12]. Two main mechanisms have been proposed to explain this result: blunting by maternal antibodies and immaturity of the infant's immune system[13]. However, despite the potential consequences for measles control—e.g., changing the recommended MCV1 age as a potential control intervention—the impact of vaccination age on vaccine effectiveness (VE) has only recently been quantified[12].

As illustrated in Fig. 1a, the increasing effectiveness of MCV1 with age should result in a trade-off in risks when recommending MCV1 age: reducing the age of vaccination increases the risk of vaccine failure while increasing the age worsens the risk of infection before vaccination. Hence, by balancing these risks, the recommended MCV1 age may be optimized to minimize measles incidence. Furthermore, location-specific factors, such as transmission level, defined here by the mean age of infection (MAI), are expected to affect this trade-off resulting in different optimal ages[14,15]. Following this conceptual model, one expects the optimal vaccination age to depend on the local epidemiology of measles.

As seen in Fig. 2, however, the global homogeneity in recommended MCV1 ages contrasts with the observed heterogeneity in measles incidence[9]. The partial Spearman rank correlation coefficient between MCV1 ages and administrative regions' mean annual incidence between 2010 and 2019 was only 0.025 ($p$-value: 0.90) in regions with ≥1 measles case per 1 million per year, when controlling for 2021 World Bank income group classification[16]. Of the 210 MCV1 recommendations obtained, 87% of recommendations were at 9 months (69 regions) or 12 months (114 regions), reflecting the recommendations from the WHO: MCV1 at 9 months in countries with ongoing transmission and at 12 months in countries with low transmission[17]. Although these recommendations generally reflect the trade-off in risks, they may not capture the complexity of factors that impact measles epidemiology, such as further transmission variation driven by differences in social contact patterns or vaccination coverage.

The relative homogeneity in MCV1 recommendations suggests opportunities for refining the MCV1 age to leverage this risk trade-off. Here, we propose a new method—based on a mechanistic model of endemic measles transmission incorporating realistic, data-driven models of MCV1 delay and VE variation with age—to estimate the optimal age to recommend MCV1. As a proof of concept, we used this method to estimate the optimal MCV1 age in a range of synthetic test populations. In these populations, we varied parameter values across realistic ranges to identify factors determining the optimal age (Fig. 1b). We show that a mismatch between the optimal and recommended age can potentially increase measles incidence. Furthermore, we show that the optimal age is sensitive to location-specific determinants of measles epidemiology, with transmission level having the greatest effect, followed by the social contact structure and vaccination coverage. Overall, our study suggests that the optimal MCV1 age is highly population-specific and, hence, more heterogeneous than the current recommendations reflect. Our findings thus suggest the potential to adjust MCV1 ages to reduce measles incidence, taking steps toward eventual elimination.

## Results

### MCV1 effectiveness increases with age of receipt
To quantify the relationship between MCV1 age and VE for inclusion in our measles model, we fitted a shape-constrained additive model[18] (SCAM) to VE estimates from a systematic review[12].

After applying our inclusion criteria (see Methods: The relationship between MCV age and vaccine effectiveness), we analyzed 53 VE estimates from 16 studies from 12 countries (see Supplementary Fig. 1 for included countries). As shown in Fig. 3a, the point estimates and the confidence intervals of VE varied greatly. Lower ages displayed particularly high variation in VE estimates. A large part of the overall variation was captured by the SCAM (59% of deviance explained), which

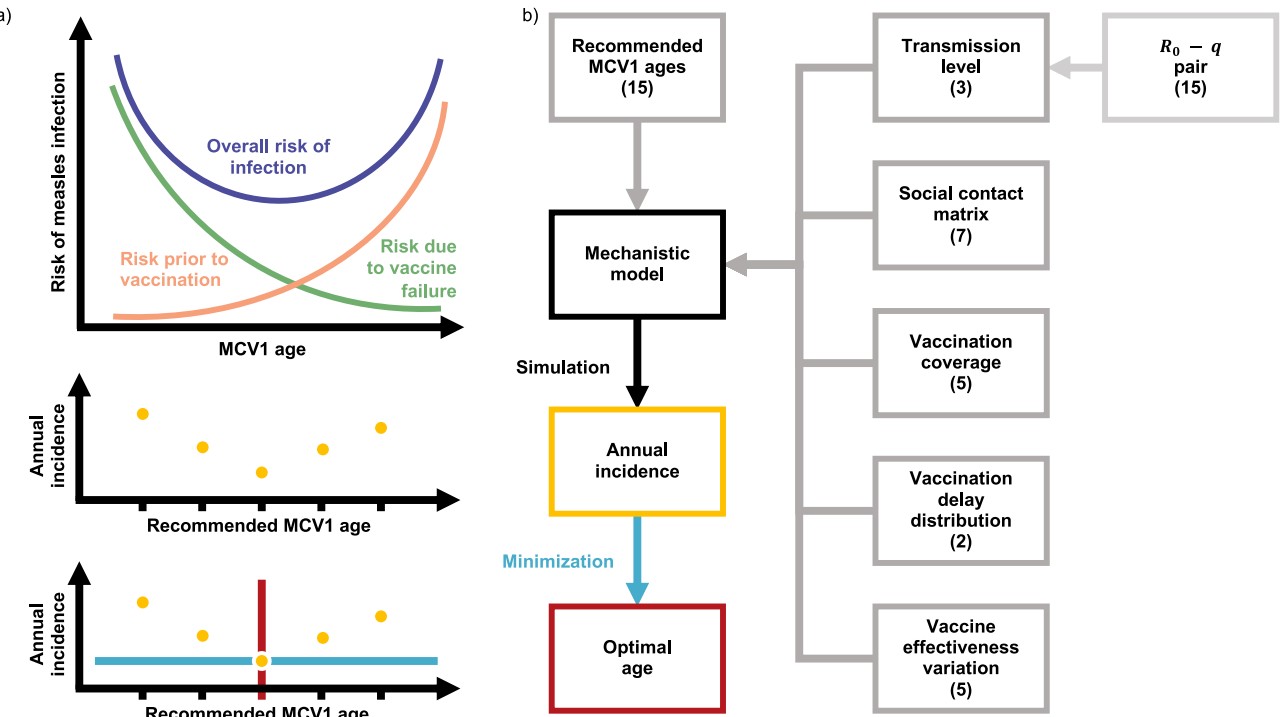

**Fig. 1 | Trade-off in risks and resulting framework for calculating the optimal age to recommend measles-containing vaccine dose 1 (MCV1). a** Illustration of the risk trade-off that should be balanced when recommending MCV1 age. **b** Framework for calculating the optimal age to recommend MCV1. Gray boxes indicate variables used to parameterize the mechanistic model of measles transmission and vaccination. Values in brackets indicate the number of variants of the variable used. The yellow dots and box reflect incidence, the blue arrow and line the minimization of the incidence, and the red indicates the age at minimum incidence: the optimal age.

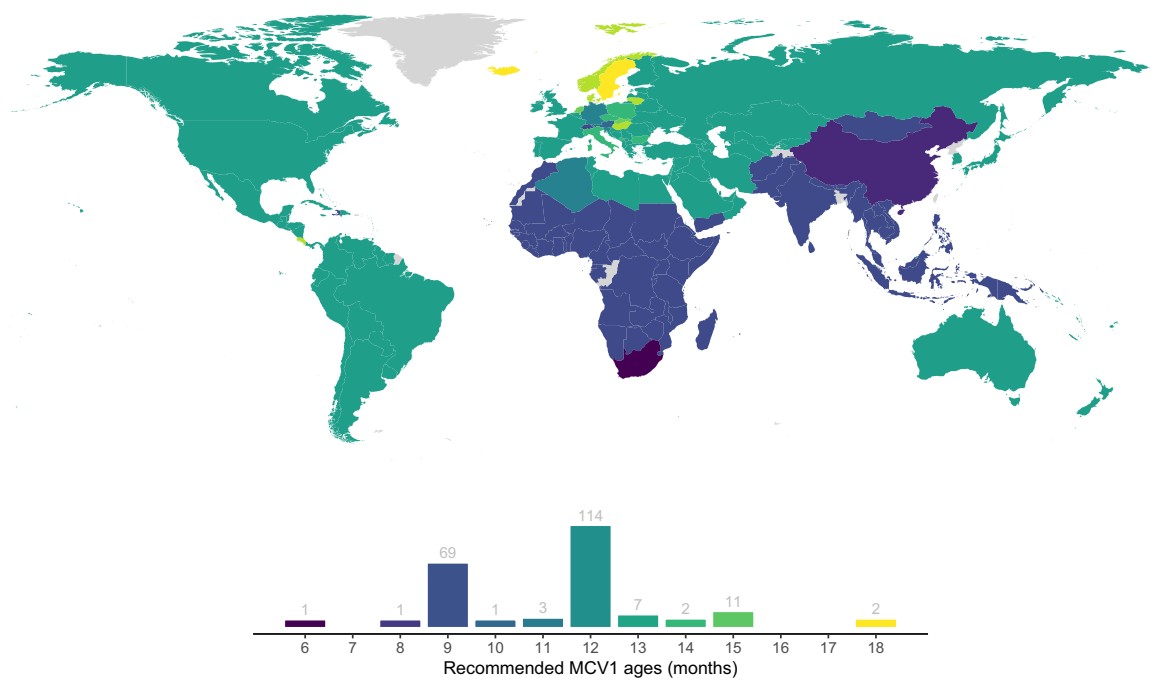

**Fig. 2 | Map and distribution of recommended measles-containing vaccine dose 1 (MCV1) ages.** Reported recommended MCV1 ages by administrative region, from the WHO[44], ECDC[45], and country-based reporting[46,47]. Regions with unknown MCV1 ages are depicted in gray. The histogram indicates the number of regions recommending MCV1 at each age. In cases where regions recommended a range of MCV1 ages (16/210 administrative regions), we report the minimum MCV1 age.

estimated an increase in VE with age, confirming the results of the earlier meta-analysis[12]. The SCAM estimated the VE at 65% at 6 months, approaching 100% by 20 months. Hence, the model confirms that, for infants ≤20 months, the effectiveness of MCV1 increases with age of receipt.

**Empirical data on the reported age of vaccination reveal that MCV1 is frequently delayed**

We quantified the distributions of MCV1 delay by fitting Lomax distributions[19] to reported 25%, 50%, and 75% delay quantiles from low- and middle-income countries[20] (see Methods: The distribution of MCV1 delay). Based on data from the 43 countries included in our analysis, we found that delays in receiving MCV1 were prevalent (median (range) of median delay: 0.6 (0.1, 1.3) months), exceeding 3 months for 25% of infants in 9 countries (median (range) of 75% delay quantile: 1.8 (0.4, 5.5) months).

In 41 of the 43 countries, we successfully fitted the Lomax distribution (see Supplementary Fig. 3), which recapitulated the 50% and 75% quantiles of the observed delay distributions. Using partitioning around medoids (PAM) clustering, we identified two broad groups of countries (Fig. 3b, Supplementary Fig. 3): one group with longer right tails (long-tailed delay, 35 countries, medoid Country: Uganda, survey median (IQR): 0.6 (0, 2.2) months, model median (IQR): 0.6 (0.2, 2.2) months) and another group with shorter right tails (short-tailed delay, 5 countries, medoid Country: Turkey, survey median (IQR): 0.7 (0.2, 1.6) months, model median (IQR): 0.7 (0.2, 1.6) months). For both clusters and all observed countries, the estimated parameters resulted in median delays in the range of 0.1 to 1.3 months, corresponding to a median delay of up to 14% when MCV1 was recommended at 9 months. Taken together, this analysis indicates MCV1 is frequently delayed, with important implications for measles control by vaccination, modeling the transmission dynamics of measles, and estimating the optimal age of MCV1.

By combining the delay distribution with age-specific MCV1 effectiveness, we calculated the cumulative effective vaccine coverage, defined as the proportion of a birth cohort protected by the vaccine by a given age (Fig. 3c). This effective coverage reflected the conceptual trade-off in risks outlined in Fig. 1a: increasing the recommended MCV1 age left a birth cohort susceptible to infection for longer but also increased MCV1 effectiveness and, hence, the long-term proportion of the cohort protected. Furthermore, the delay distribution also determined the effective coverage, with longer delays resulting in lower proportions of a birth cohort protected 18 months after the recommended age (Fig. 3d).

**Heterogeneity in transmissibility is necessary to recapitulate pre-vaccine reports of measles MAI**

To calibrate our model (described in Methods: Model of measles transmission and vaccination, and the supplementary material), we parameterized the model's inter-age-group contacts using seven social contact matrices[21,22] (SCM). Based on MAI values observed in the pre-vaccine era[23], we defined three transmission levels: high (24–36 months), medium (36–48 months), and low (48–72 months). For each SCM and transmission level, we calibrated $R_0$ and $q$ (the transmissibility of <5-year-olds relative to the rest of the population) to match the target pre-vaccine MAIs (see Methods: Recapitulating reported pre-vaccine mean ages of measles infection).

When comparing model-derived MAIs to historical estimates in the pre-vaccine era, we found that, for typically reported values of $R_0$[24] between 12 and 18, the model failed to recapitulate the MAIs for multiple SCMs (Supplementary Fig. 8). However, once age-specific transmissibility was included, the transmission model could recapitulate all historical estimates of MAIs for all SCMs, except the China SCM for the MAI of 24 months. All fitted values of $q$ and $R_0$ converged according to the convergence criteria (described in Methods: Recapitulating reported pre-vaccine mean ages of measles infection). These fitted pairs displayed a negative association, such that increases in $R_0$ were compensated by decreases in $q$. Hence, this calibration allowed us to define parameter regions that reproduce each transmission level (Fig. 4) for inclusion in the transmission model.

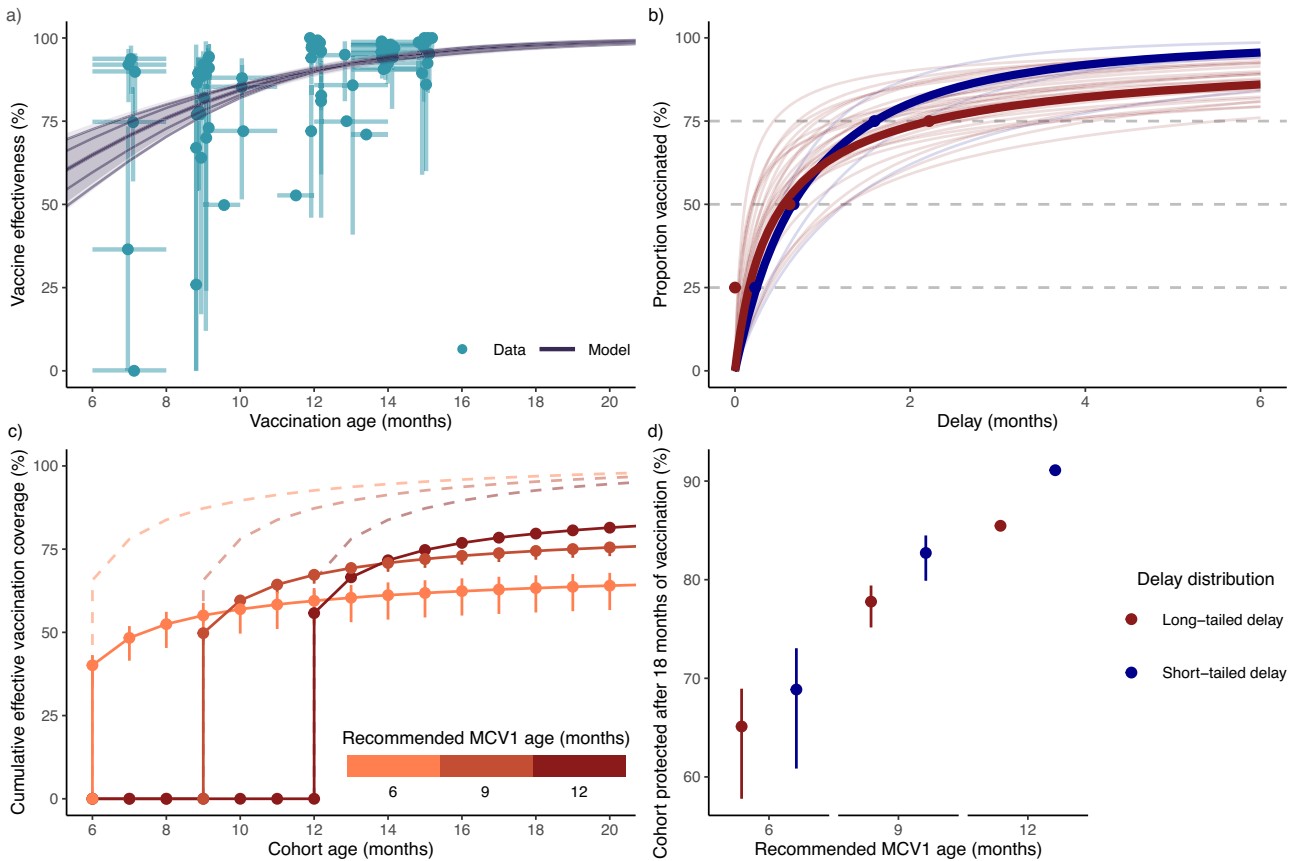

**Fig. 3 | The impact of recommended measles-containing vaccine dose 1 (MCV1) age on the effective vaccine coverage. a** MCV1 effectiveness with the age of receipt. Points represent the reported measure of vaccine effectiveness (VE) estimates from a systematic review[12] (*y*-axis) at the MCV1 age range midpoint (*x*-axis), with vertical lines indicating the VE estimate 95% confidence interval and horizontal lines indicating the MCV1 age range. Fifty-three VE estimates are presented. Sample sizes for each estimate are detailed in Supplementary Data 2 of the data source[12]. The shape-constrained additive model (SCAM) is indicated in gray; the shaded area indicates the approximate simultaneous 95% confidence interval, and the lines indicate the 2.5%, 25%, maximum likelihood estimate (MLE), 75%, and 97.5% quantiles. **b** Cumulative delay distributions. The delay distributions were modeled using the Lomax distribution, a two-parameter distribution extension of the Exponential distribution with heavier right tails. Cluster medoids are bolded, with points showing the associated delay data. **c** Cumulate effective MCV1 coverage when recommending MCV1 at 6, 9, and 12 months, with long-tailed delay distribution. Cumulative vaccine coverage after 24 months is set to 100%. Dashed lines indicate the MCV1 coverage, points indicate the MLE effective MCV1 coverage, and vertical lines indicate the 95% confidence intervals resulting from the 95% confidence interval of the SCAM. **d** Cumulative effective MCV1 coverage after 18 months when recommending MCV1 at 6, 9, and 12 months, for long-tailed and short-tailed delay distributions. Points indicate MLE estimates and vertical lines indicate the 95% confidence intervals resulting from the 95% confidence interval of the SCAM.

## The optimal age to recommend MVC1 is sensitive to transmission level, contact structure, and vaccine coverage

To estimate the optimal age to recommend MCV1, we simulated annual measles equilibrium incidence. We simulated recommending MCV1 at ages between 6 and 20 months and identified the optimal recommended age, corresponding to the minimum incidence (see Methods: The optimal age to recommend MCV1). To explore the possible range in optimal ages, along with factors that affect the optimal age, we calculated the optimal age for populations with all possible combinations of parameters (see Fig. 1b), varying the SCM (China, India, Japan, Moscow, South Africa, UK, USA), MCV1 coverage (45%, 55%, 65%, 75%, 85%), transmission level (low, medium, high) defined by the $R_0-q$ pairing, VE curve (2.5%, 25%, maximum likelihood estimate (MLE), 75%, 97.5%), and delay distribution (short-tailed delay, long-tailed delay).

Of all the parameter sets modeled, 99.6% fulfilled our convergence criteria. Across those, we identified a unique optimal age ranging from 6 to 20 months between scenarios. Furthermore, the predicted optimal ages varied greatly between scenarios (see Fig. 5a, b). For example, the optimal age for the low-transmission level with the China SCM at 85% MCV1 coverage ranged from 16 to 20 months, whereas the high-transmission level at 45% vaccine coverage with the South Africa SCM ranged from 6 to 9 months. Moreover, recommending a non-optimal MCV1 age could result in up to a 2.4-fold increase in incidence, although the impact of an age mismatch was similarly scenario-dependent (see Fig. 5a, Table 1).

Of the factors we varied, the transmission level impacted the optimal age the most. At 45% MCV1 coverage, parameterized with the USA SCM, the optimal age ranged between 12 to 15 months in a low-transmission setting and 8 to 11 months in a high-transmission setting. More generally, increasing transmission from low to medium decreased the optimal age by an average of 1.7 months, and increasing from low to high transmission resulted in an average decrease of 3.7 months (Supplementary Table 4). Higher transmission resulted in lower MAIs and increased the risk of infection at younger ages, thus resulting in younger optimal ages to compensate for this risk. Accordingly, increases in the pre-vaccine MAIs resulted in increases in the optimal age (Supplementary Fig. 9a).

Social contact structure also affected the optimal age to recommend MCV1. Even after controlling for transmission level and vaccine coverage, the range of optimal ages varied between SCMs:

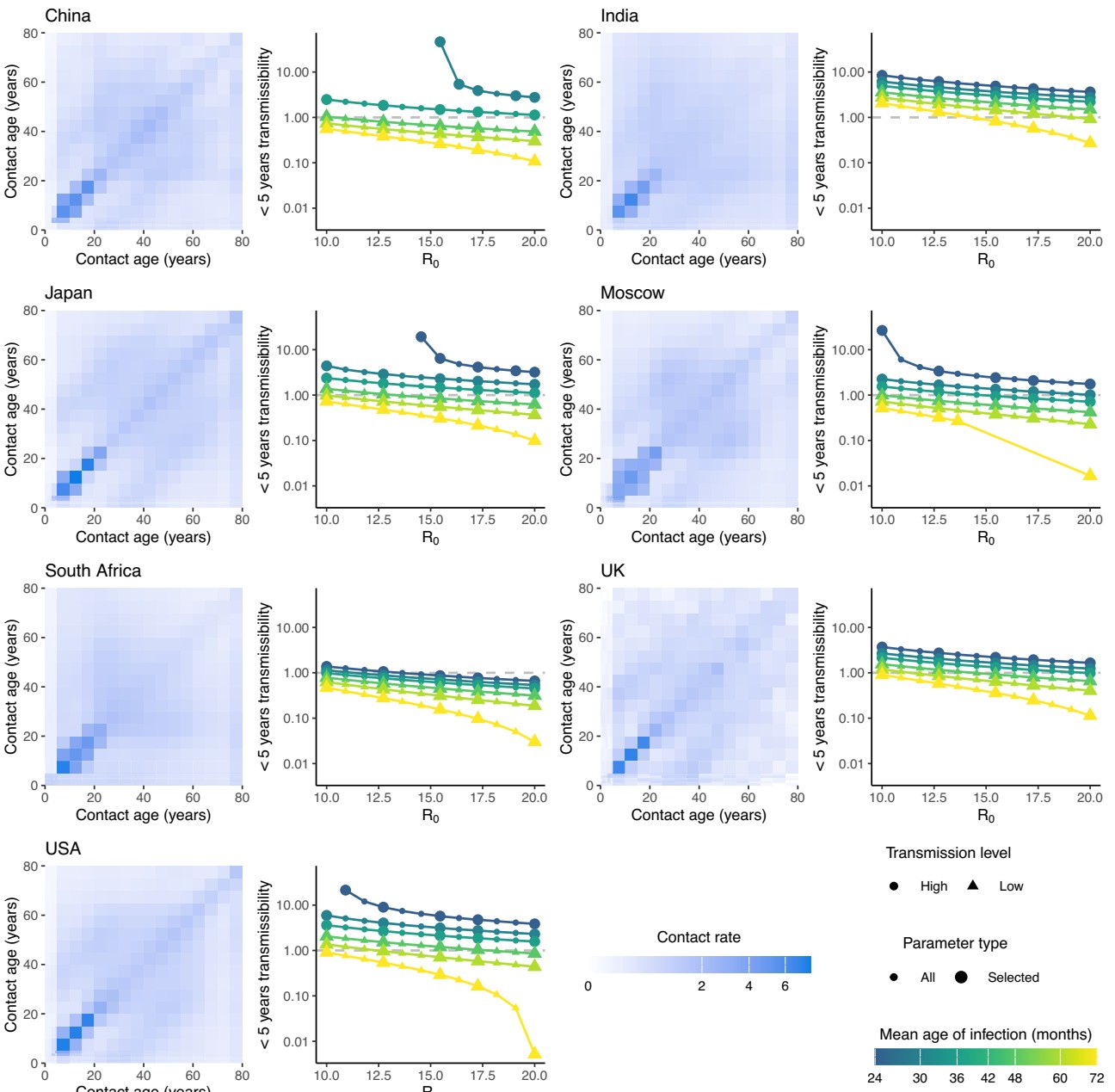

**Fig. 4 | Basic reproductive number and relative transmissibility in <5-year-olds for 7 social contact matrices (SCM).** For each SCM, the left-hand plots show the social contact rates between age groups. Color intensity indicates the per-capita daily contact rate of the x-axis age group with the y-axis age group. In the corresponding right-hand plots, points indicate the fitted values of $R_0$ and $q$ for each target mean age of infection, with larger points indicating the values selected to parameterize the mechanistic model in further analysis. Transmission levels are indicated by point shape, with the mean age of infection denoted by point color. For clarity, medium-transmission level points are not shown.

for example, from 6 to 9 months for the South Africa SCM to 10 to 13 months for the China SCM for a scenario with high transmission at 45% MCV1 coverage. Depending on the MCV coverage–transmission level scenario considered, the optimal ages clustered into 2 to 5 groups. However, ≥3 groups were typically required to capture the heterogeneity in optimal ages between SCMs (13/15 scenarios, Supplementary Fig. 10). In most scenarios, the China SCM and the South Africa SCM tended to cluster independently, representing the SCMs with the oldest and youngest optimal ages respectively, with other SCMs clustering together, corresponding to optimal ages between these groups.

Finally, vaccine coverage also impacted the optimal age. Specifically, increased coverage reduced measles incidence, resulting in

higher optimal ages. For example, optimal MCV1 ages in a low-transmission setting with the South Africa SCM ranged from 9 to 13 months at 45% vaccine coverage and from 11 to 16 months at 85% vaccine coverage. In general, when accounting for transmission level and SCM, a 10 percentage point increase in MCV1 coverage resulted in an average increase in optimal age of 0.6 months (Supplementary Table 5). Overall, these results demonstrate the importance of location-specific factors of measles epidemiology for vaccine policy.

## The impact of variations in age-specific MCV1 effectiveness and delay distribution on the optimal age is minor

Uncertainty in the vaccine effectiveness curve had a comparatively reduced effect on the optimal age. When holding all other

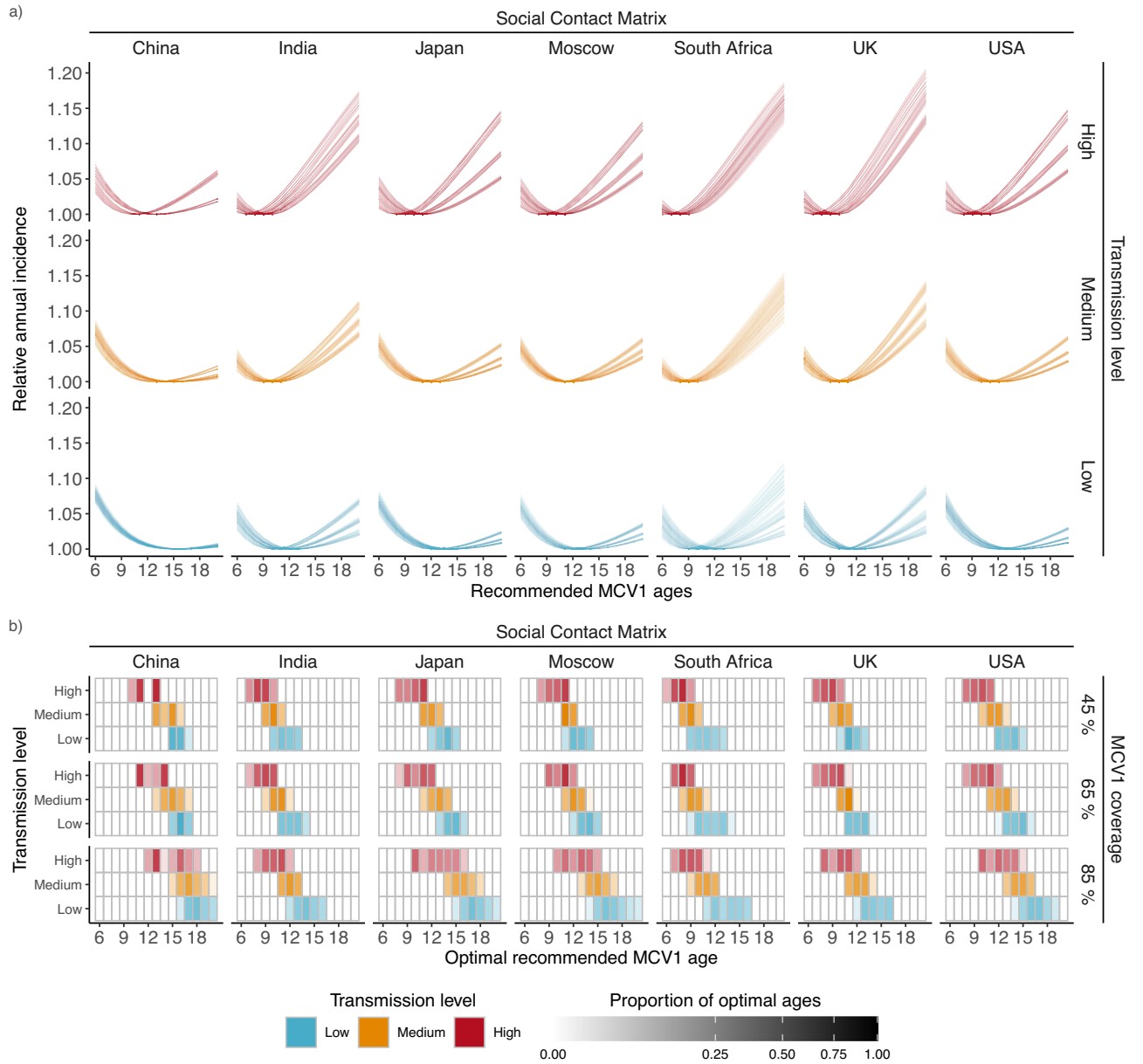

**Fig. 5 | Estimating the optimal age to recommend measles vaccination.**
**a** Estimated annual incidence when recommending measles-containing vaccine dose 1 (MCV1) at ages 6 to 20 months, with 45% MCV1 coverage. Each line indicates the incidence for a given parameter set relative to the minimum incidence for that parameter set. **b** Heatmap of optimal ages. Opacity indicates the proportion of parameter sets with an optimum at a given MCV1 age. For clarity, the 55% and 75% MCV1 coverage results are not displayed.

parameters constant, varying the curve resulted in changes in the optimal age in 56.7% of parameter sets. In cases where the optimal age varied, the effect was inconsistent, but generally, the optimal age increased as the quantile of the VE curve increased (Supplementary Fig. 9b).

Uncertainty in the MCV delay distribution had only a minor impact on optimal age. When holding all other parameters fixed, changing the delay distribution resulted in a change in optimal age in only 12.2% of parameter sets. When changes occurred, a higher optimal age was predicted for the short-tailed delay distribution (Supplementary Fig. 9c).

### The optimal age is highly similar across lower-income social contact matrices

To better recapitulate high-burden settings, we estimated the optimal MCV1 ages at high transmission level for 4 SCMs from low- and lower-

middle-income countries[25] (LMICs)(Ghana, Sierra Leone, Uganda, and Zambia), as defined by the 2021 World Bank income groups[16] (see Methods: Optimizing the MCV1 ages for social contact matrices from low- and lower-middle-income countries). All simulations passed our convergence criteria, allowing us to estimate an optimal age for every parameter combination (range 7–11 months). The SCM had only a minor impact on the optimal MCV1 age, with only slight variations in ranges between scenarios (see Fig. 6). This is potentially due to the high degree of similarity in the SCMs (mantel test >0.99 for all LMIC SCM pairs, between type 1 demography assortativity-corrected SCMs, see Supplementary Fig. 12). However, this similarity could be artificially introduced by the low age-group resolution from the LMICs SCMs. Furthermore, the variation in the MCV1 coverage also had a reduced impact on the optimal age. While increased coverage still led to older optimal ages, this increase was lower than for the higher resolution SCMs presented above, with a 10 percentage point increase

**Table 1 | Estimated optimal ages and annual incidences**

| SCM | Optimal age | Incidence at the optimal age | 9 months | | 12 months | | |
|---|---|---|---|---|---|---|---|
| | | | % Increased incidence | Incidence rate difference | % Increased incidence | Incidence rate difference | |
| China | 12 (12, 12) | 757 (753, 761) | 1 (1, 1) | 7 (7, 8) | 0 (0, 0) | 1 (1, 1) | High transmission, 45% MCV1 coverage |
| India | 8 (8, 9) | 851 (847, 856) | 0 (0, 0) | 2 (1, 2) | 2 (2, 2) | 19 (18, 21) | |
| Japan | 10 (10, 10) | 803 (797, 808) | 0 (0, 0) | 2 (2, 3) | 1 (1, 1) | 8 (7, 9) | |
| Moscow | 10 (10, 10) | 800 (795, 804) | 0 (0, 0) | 2 (2, 2) | 1 (1, 1) | 7 (6, 7) | |
| South Africa | 8 (8, 8) | 881 (876, 885) | 0 (0, 1) | 4 (3, 5) | 3 (3, 4) | 30 (28, 31) | |
| UK | 8 (8, 9) | 842 (836, 847) | 0 (0, 0) | 2 (2, 3) | 3 (2, 3) | 22 (21, 24) | |
| USA | 10 (9, 10) | 817 (812, 822) | 0 (0, 0) | 2 (1, 2) | 1 (1, 1) | 9 (8, 11) | |
| China | 18 (18, 18) | 171 (167, 175) | 21 (20, 22) | 34 (33, 35) | 7 (6, 7) | 11 (10, 11) | Low transmission, 85% MCV1 coverage |
| India | 14 (14, 14) | 210 (206, 213) | 11 (10, 11) | 22 (21, 23) | 2 (1, 2) | 3 (3, 3) | |
| Japan | 17 (17, 18) | 175 (171, 179) | 20 (18, 21) | 32 (31, 33) | 6 (5, 6) | 10 (9, 10) | |
| Moscow | 17 (16, 17) | 176 (172, 181) | 19 (18, 20) | 31 (30, 32) | 5 (5, 6) | 9 (8, 9) | |
| South Africa | 14 (13, 14) | 211 (206, 215) | 10 (9, 11) | 20 (19, 21) | 1 (1, 2) | 3 (2, 3) | |
| UK | 14 (14, 15) | 194 (190, 198) | 13 (12, 14) | 24 (23, 25) | 2 (2, 2) | 4 (3, 4) | |
| USA | 17 (16, 17) | 183 (179, 187) | 17 (17, 18) | 31 (30, 32) | 5 (4, 5) | 8 (8, 9) | |

Optimal age indicates the mean (95% confidence interval) of the optimal ages from all parameter sets for the social contact matrix (SCM) and measles-containing vaccine dose 1 (MCV1) coverage scenario. Incidence indicates the mean (95% confidence interval) annual incidence per 100,000 across parameter sets at the optimal ages. Percentage increased incidence reflects the mean (95% confidence interval) of the incidence increase compared to the optimal age incidence. Incidence rate difference quantifies the mean (95% confidence interval) absolute difference in annual measles incidence per 100,000 compared to the optimal age.

in MCV1 coverage resulting in an increase in optimal age of 0.3 months when accounting for SCM (see Supplementary Table 6).

**Changes to the demographic structure have a substantial impact on the incidence but only a minor impact on the optimal MCV1 ages**

To further increase similarity to high measles burden countries, we varied the population demography between type 1 and type 3 survival[26]. Varying the demographic type had a substantial impact on the simulated incidence (see Fig. 6c), with type 3 demography resulting in a mean optimal annual incidence of 2,262 (95% CI: 2254, 2270) per 100,000 compared to type 1 demography which resulted in an annual incidence of 866 (95% CI: 864, 868) across all simulated scenarios at 45% MCV1 coverage. Furthermore, the impact of demographic type decreased with MCV1 coverage, with 85% MCV1 coverage resulting in type 3 demography annual incidence of 732 (95% CI: 722, 743), and type 1 in an annual incidence of 322 (95% CI: 319, 326).

A comparison of the age-specific incidence (see Supplementary Fig. 13) reveals this difference in incidence is driven by the difference in population sizes in younger age groups. When comparing age-specific incidences, the difference between demographic groups becomes minor. As we model transmission as frequency-dependent, this is as expected. Furthermore, despite significant differences in incidence, the demographic type had no significant impact on the optimal age. When moving from type 1 to type 3, the optimal age decreased by 0.016 months (95% CI: −0.058, 0.027) when controlling for MCV1 coverage and SCM (see Supplementary Table 7).

**The relative transmissibility parameter cutoff has minor impacts on the optimal age**

To assess the impact of the cutoff age for the relative transmissibility parameter, we estimated the optimal age for the South Africa SCM with a cutoff of 3 years of age. When parameterized with $q$ having a cutoff at 3 years, the model successfully recapitulated all the target MAIs. Furthermore, we were able to find the optimal MCV1 age for each scenario with this alternative $q$ parameterization (see Supplementary Fig. 6). Comparing the alternative $q$ parameterization to the 5-year cutoff, we found the estimated optimal ages were similar (5-year range: 6–16 months, 3-year range 6–18 months). Moreover,

the results remain qualitatively the same: increased transmission resulted in a reduced optimal age, with increased vaccine coverage leading to a corresponding increase in optimal ages. However, variation in the cutoff age led to variation in the optimal ages, highlighting the importance of tailoring our measles transmission model based on detailed measles data in the study population to accurately determine the optimal age.

**The timescale for changes to take effect reduces with increased transmission**

We estimated the time to converge to within 10% of the optimal incidence when changing to the optimal age from an initial MCV1 age of either 9 or 12 months (see Supplementary Fig. 14). Overall, the median time to converge was 4.0 years after the schedule change (IQR: 0.0, 12.1 years). However, as expected from epidemiological theory[27], the convergence time was highly dependent on transmission, with lower transmission levels leading to longer times to converge (low: 9 years (IQR: 2.5, 27.7 years), medium: 3.9 years (IQR: 0.0, 11.0 years), high: 2.2 years (IQR: 0.0, 4.5 years). High variability in convergence times was also observed across the SCMs, with the SCM for South Africa corresponding to the shortest median convergence period (0 years, IQR: 0.0, 6.3 years), and the SCM for China corresponding to the longest median convergence period (7.5 years, IQR: 2.6, 18.5 years).

## Discussion
### Summary
In this study, we developed a new method to estimate the optimal age to recommend MCV1 based on a mechanistic model of measles transmission and vaccination. In particular, this model captured several complexities of measles epidemiology, namely age-specific contacts, vaccination delays, and VE variations with age of receipt. For every scenario tested, we could identify a unique optimal age in the range of 6 to 20 months, contrasting the two ages recommended by the WHO. Furthermore, we found that the optimal age was governed by location-specific factors, namely transmission level, vaccine coverage, and social contact structure. Overall, our results suggest that in addition to increasing vaccine coverage, adjusting the recommended vaccination age could help minimize immunity gaps and reduce measles incidence, taking steps toward eventual elimination.

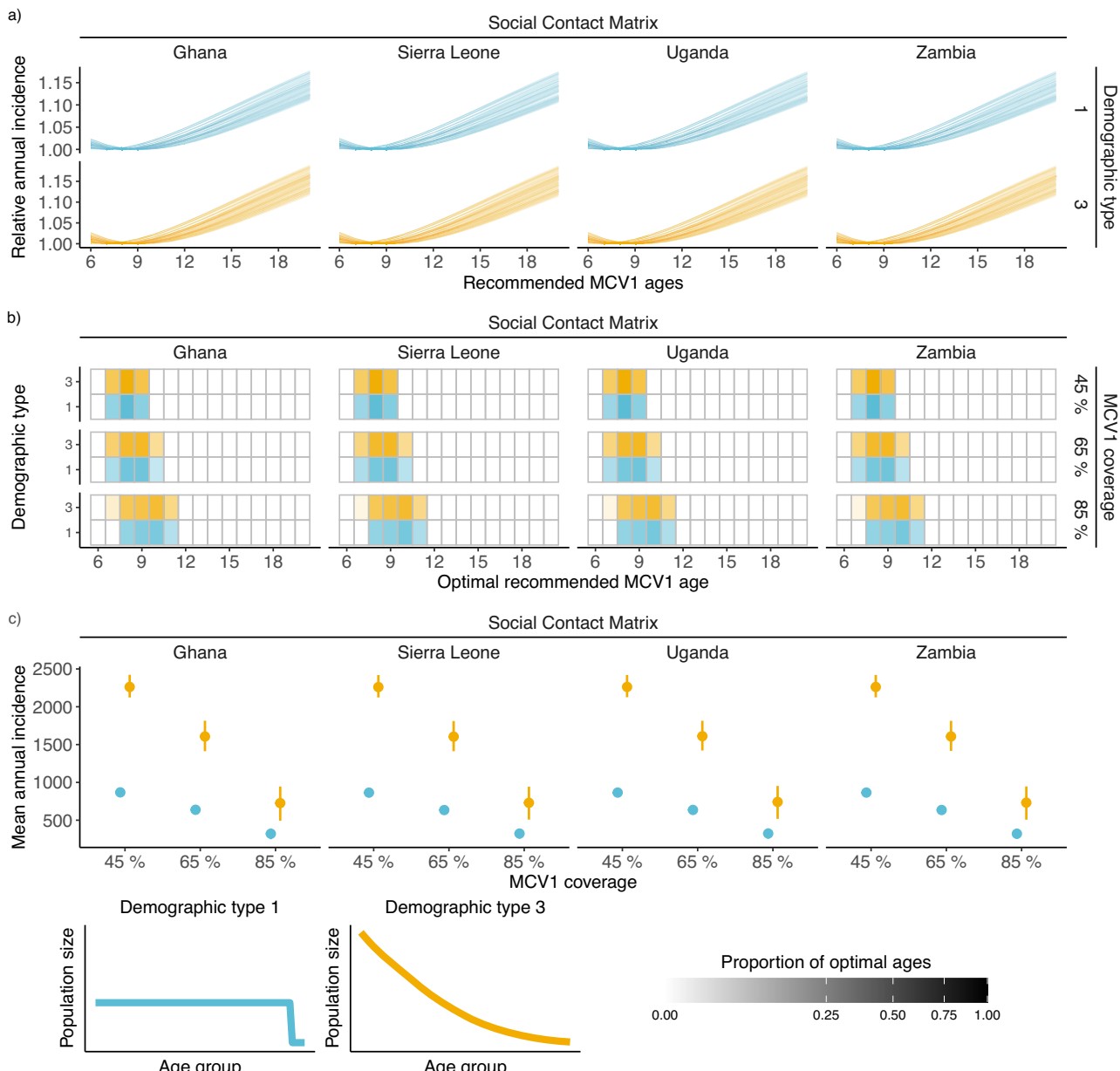

**Fig. 6 | Estimating the optimal measles-containing vaccine dose 1 (MCV1) age in high-burden scenarios. a** Estimated annual incidence when recommending MCV1 at ages 6–20 months, with 45% MCV1 coverage. Each line indicates the relative incidence for a given parameter set relative to the minimum incidence for that parameter set. **b** Heatmap of optimal ages. Opacity indicates the proportion of parameter sets with an optimum at a given MCV1 age. For clarity, the 55% and 75% MCV1 coverage results are not displayed. **c** Mean annual incidence per 100,000 at the optimal MCV1 age by demographic type for different MCV1 coverage. Points indicate the mean incidence, and lines indicate the 2.5% and 97.5% quantiles.

## Impact of local transmission parameters on the optimal age

A key result of our study is that the optimal MCV1 age depends on the local epidemiology of measles. Specifically, we predict that populations with high measles transmission require a lower vaccination age. More generally, the impact of some influential factors of local epidemiology on the optimal age can be understood by considering their effect on the MAI and transmission after vaccine introduction. This explains the effect on the optimal age of pre-vaccine transmission levels (controlled by the parameters $R_0$ and $q$), which correlate with post-vaccination transmission levels (Spearman rank correlation between pre- and post-vaccination MAI: 0.76). Similarly, increasing vaccination coverage is dynamically equivalent to reducing transmission[28], resulting in older optimal ages.

When comparing simulated MAI to observed MAI, two transmission parameters were required to recapitulate the range of transmission intensities observed in the pre-vaccine era. Unlike earlier modeling studies[29,30], varying only the basic reproduction number was insufficient to reach all target MAIs for all SCMs. This discrepancy may be explained by the high age resolution and the inclusion of realistic SCMs in our model. Moreover, this result suggests age heterogeneities beyond social contacts are necessary for the design of realistic models of measles transmission. Here, we allowed the relative transmissibility of <5-year-olds to vary, and calibrated values varied across several orders of magnitude (range: 0.005–46.3). Hence, the heterogeneity in the relative transmissibility parameter could be interpreted as an adjustment to the SCM, either because the SCMs used were derived for more modern populations than the pre-vaccine MAI estimates, or due to systematic biases resulting from diary-based reporting of social contacts[22]. However, as demonstrated by our simulations using an alternative relative transmissibility cutoff (3 years old) other definitions

of this parameter are possible, and the choice should be based on age-specific factors affecting measles in the study population.

## Impact of social contact structure on the optimal age

Even for fixed transmission levels and vaccination coverages, the impact of different social contact structures was strong. This impact demonstrates that, in addition to quantifying the transmission of measles, detailed knowledge of social contact structure, quantified by data-driven SCMs, is needed to identify the optimal age in a given population. In our analysis, we used three sources of SCM. The first[22] generated SCMs from diary-based reporting for limited locations, while the second[21] and third[25] SCMs sources either fit to or extrapolated from the diary-based SCMs to infer SCMs for more locations, hence inheriting the limitations inherent to diary-based SCMs alongside method-specific issues. The second SCMs source[21] resulted in highly resolved age-specific contact rates but only applied the method in a low number of LMICs. The third source[25] had better coverage of LMICs but a lower age resolution, potentially masking differences between countries' social contact structures and optimal ages. Overall, this indicates the need for highly resolved data-driven SCMs, particularly for regions with the greatest burden of infectious diseases, namely LMICs, to more accurately identify the optimal MCV1 ages.

## Impact of demographic structure on the optimal age

Our simulations in synthetic LMIC populations included comparing the impact of demography on the optimal age. Our analysis showed that the optimal age was unchanged by demographic structure. This result implies that changes to demographic structure with time would not impact the optimal age. However, this result should be interpreted carefully, as we only tested type 3 demography with LMICs SCMs and only considered demographic types 1 and 3. Furthermore, we assumed frequency-dependent transmission, but other assumptions may cause demographic differences to have a larger effect on the optimal age. Hence, more work will be needed to assess the impact of demography.

## Limitations and further requirements for applying the method to real-world settings

We chose the minimization endpoint of equilibrium incidence to estimate the optimal age in endemic settings, as the primary goal of our study was to establish a proof of concept. In real-world applications, however, this endpoint should be reassessed frequently as the optimal ages change with decreasing transmission. Our simulations suggest a decade would be an appropriate interval to reassess the optimal age (median time to converge to the optimal incidence 4.0 years (IQR: 0.0, 12.1)), although these simulations used the equilibrium endpoint. Additionally, other endpoints like hospitalizations or deaths may be considered, but they will require extending our model to represent additional mechanisms of vaccine protection, such as reduced disease severity in vaccine breakthrough cases[11]. Importantly, considering other endpoints may change the nature of the risk trade-off, for example, because of the increased incentive to vaccinate younger if mortality is increased in younger ages[31]. Similarly, alternative endpoints, like the susceptible population size[32] or the risk of invasion, will be needed to estimate the optimal age in elimination settings, where vulnerability to outbreaks may persist due to residual pockets of susceptible individuals[33]. Applying our model in such settings will require a stochastic formulation due to the low number of cases and frequent extinctions that deterministic models fail to capture well.

Furthermore, the real-world application of our method will require additional components beyond population-specific information on the SCM, vaccine coverage, and delay distribution to fully characterize the measles epidemiological dynamics in a target population. These include detailed information on past and current MCV1 and MCV dose 2 (MCV2) coverages to build up a detailed picture of the population's current immune status. A second key component is transmission seasonality, which can result from term-time increases in contacts among school-aged children[27] or the effect of climate on virus transmissibility[34]. Furthermore, as shown by our sensitivity analysis, changing the cutoff age of the relative transmissibility parameter impacts the optimal age. Hence, the model structure requires careful consideration. Therefore, a prerequisite to applying our method is a detailed model—identified, for example, by fitting to long-term incidence data using modern statistical inference techniques[35]—for capturing the local drivers of measles transmission.

We recognize, however, that the application of the method has high data and modeling capability requirements, which may make the application challenging in data-poor areas. While making recommendations based on our current synthetic simulations would be inappropriate, future work based on our method may allow for more generalizable recommendations. For example, systematic application of our method to real-world populations for various SCMs, vaccine coverages, and measles risk levels would illuminate the range of realistic and observable optimal ages. Alternatively, as shown in Supplementary Fig. 11, we find a positive relationship between the optimal age and the post-vaccination MAI, even when vaccinating at non-optimal ages. Hence, once the method has been applied to sufficient populations, it may be possible to make recommendations based on the post-vaccination MAI, MCV1 coverage, and social contact structure. Furthermore, as previously shown[21], SCMs are generally similar across geographical regions; hence, it may be possible to make regional recommendations. This would allow countries to assess the appropriate MCV1 schedule relatively easily, without losing all the heterogeneity in optimal ages identified in our analysis.

Another key consideration when applying our proposed method is SIAs. These additional immunization campaigns aim to rapidly increase population immunity by vaccinating target demographics—typically children aged ≤14 years—regardless of vaccination history[36]. In general, such campaigns are expected to reduce transmission and, thus, increase the optimal age for MCV1. Hence, in settings where SIAs are routinely administered, MCV1 age should be optimized to maximize the effect of both interventions.

We consider changing the recommended MCV1 age to be a relatively low-cost intervention when compared to trying to increase MCV1 coverage. As MCV1 is already part of the vaccine schedules in all countries, changes to the MCV1 age could lead to changes in measles incidence without requiring changes to MCV1 coverage. However, changing the recommended MCV1 age would still require considerable effort and consideration of impacts. For example, changes to the MCV1 age could leave cohorts unprotected if they are not covered by either the old or new schedules. In this case, additional immunization activities would be necessary to prevent immunity gaps from emerging. Furthermore, many countries group vaccine schedules together. Hence, moving the MCV1 away from the co-scheduled vaccine doses may lead to decreased vaccine uptake. Because our method quantifies the incidence at potential MCV1 ages, it is possible to estimate the difference in incidence from using a non-optimal MCV1 age and consider whether the optimal age is the best policy choice.

The potential trade-off between the optimal MCV1 age and maximizing MCV1 coverage will need to be carefully considered. As our experiments in synthetic populations have shown, changes to the MCV1 coverage have a greater impact on annual incidence than changes to the MCV1 age. Hence, maximizing vaccine coverage should remain a priority for public health agencies. In light of this, the relationship between vaccine hesitancy and vaccine schedules should be considered when choosing the recommended MCV1 age. A subset of vaccine-hesitant parents choose not to forgo vaccination entirely but rather delay doses[37]. Hence, if older MCV1 ages lead to increased coverage, then lower incidence may be achieved by recommending a non-optimal MCV1 age to maximize coverage.

## Outlook

Beyond optimizing the optimal age to minimize measles incidence, our method could be extended to consider the effects of MCVs on other pathogens. Indeed, measles infection can cause immune amnesia, whereby the suppression of immune cells partially erases immune memory to previously encountered pathogens[38]. As a result, MCVs have beneficial indirect effects on other infectious diseases[39]. In addition, it has been proposed that MCVs directly affect non-measles pathogens through trained immunity[40]. Beyond MCVs, our proposed method could also be applied to estimate the optimal age for vaccination against other childhood infections. We expect the relationship between vaccination age and VE to be qualitatively similar for other vaccines, though the empirical evidence remains more limited than for measles[41,42]. As the measles vaccine is often combined with the mumps and rubella vaccines (MMR), the natural next candidates would be these two vaccines. Because mumps and rubella have different transmissibility than measles (Mumps $R_0$: 4–7[43], Rubella: 6–7[43]), the risk trade-off underlying the optimal age is expected to differ. Hence, a future research question is how to extend our approach to identify the optimal age for combined vaccines.

Despite the availability of effective vaccines for over 60 years, measles remains a considerable threat in many countries. Here, we propose that, alongside ever-necessary efforts to increase vaccine coverage, another effective intervention to reduce measles cases may be to tailor the vaccination age. Hence, our results suggest the scope for public health authorities to improve measles control and reach for eventual elimination by customizing the recommended vaccination schedule. More generally, as the trade-off underlying the optimal age is not specific to measles, our results could have ramifications for controlling many other vaccine-preventable diseases.

## Methods

### Data on measles incidence recommended MCV1 age and income group classification

We gathered data on MCV1 recommended ages from the WHO[44] and the European Center for Disease Prevention and Control (ECDC)[45], supplemented with reports from individual countries in cases of missing data[46,47]. In cases where a range of MCV1 ages was recommended, we extracted the minimum age. We also collected country-level estimates of annual measles incidence for 2010–2019 from the WHO[9] and 2021 income group classification from the World Bank[16].

### The relationship between MCV age and vaccine effectiveness

To quantify how vaccination age affects MCV VE, we fitted a statistical model to reported estimates of MCV VE, obtained from a systematic review[12] (Supplementary Data 2[12]). For MCV1 VE, to reduce uncertainty in MCV1 age, we only included estimates with an MCV1 age interval of <3 months. For the included MCV1 VE estimates, we calculated each estimate's standard error from reported 95% confidence intervals[48].

We then fitted a SCAM[18] to the logit-transformed MCV1 VE estimates. Specifically, we used a monotonically increasing P-spline basis with 4 knots, weighted according to precision, with standard errors transformed using the Delta method. We then calculated approximate simultaneous confidence intervals to assess uncertainty in model fit[49]. To include this uncertainty in the transmission model, we considered 5 curves corresponding to the predicted 2.5%, 25%, 75%, and 97.5% quantiles and the maximum likelihood estimate (MLE) from the SCAM. The uncertainty in the MCV1 VE relationship may reflect both the variation between infant immune system maturation and in maternal antibody blunting levels. Hence, we can capture the impact of both biological mechanisms by sampling the breadth of uncertainty in the MCV1 VE relationship.

For MCV2 VE, too few estimates (17 estimates at 3 unique MCV2 ages with age range <3 months) were available to assess age dependencies (Supplementary Fig. 2). We, therefore, assumed MCV2 VE to be constant and equal to the mean MCV2 VE.

### The distribution of MCV1 delay

To incorporate realistic distributions of MCV delay (i.e., the delay between the recommended age and the actual age of administration), we obtained data on MCV1 delay between 1996 and 2005 in 45 low- and middle-income countries[20], where most measles deaths occur[6]. The data consisted of the observed 25%, 50%, and 75% quantiles of the delay distribution. We excluded two countries (Colombia and the Dominican Republic) from the analysis, as both countries' delay distributions were affected by changes to the vaccination programs in response to local measles outbreaks[20] (see Supplementary Fig. 3 for included countries). We initially fitted an Exponential distribution, which failed to capture the observed long right tails. Hence, we then fitted a Lomax distribution[19] (an extension of the Exponential distribution with longer right tails to capture long delays) to every country by minimizing the squared distance between the simulated and observed quantiles. To summarize the variation in delay distributions, we clustered the Lomax distribution parameters using PAM Euclidean distance clustering[50]. The number of clusters was determined using the average silhouette method[50] and the Gap statistic[51], with 500 bootstraps.

### Model of measles transmission and vaccination

To simulate measles incidence when recommending MCV1 at different ages, we constructed a mechanistic model of measles transmission and vaccination, incorporating the aforementioned data-driven statistical models of MCV1 delay and age-specific MCV1 VE. The model was a deterministic SIR model, which split the population by infection status into susceptible, infectious, recovered, and protected by maternal antibodies. For sufficiently large populations, deterministic models have been shown to capture the dynamics of measles[52,53].

The model was age-structured, to allow the vaccination age to vary. The model split the population into monthly age groups between ages 0 and 59 months, then into 5-year age groups between ages 5 and 79. We assumed type 1 survivorship[26] (i.e., all individuals achieve the population life expectancy, then die), corresponding to equally sized 5-yearly age groups, and constant overall population size. Contacts between age groups were parameterized using data-derived SCMs[21,22]. To capture the variability in social contact structure, we selected 7 SCMs with high age group resolution (yearly age groups) derived from China, India, Japan, Moscow, South Africa, the UK, and the USA, representing the clusters identified in a previous study that clustered SCMs from 35 countries and 277 subnational administrative regions[21].

To model vaccination with two doses of MCV, we added a vaccinated susceptible state to model infants with primary vaccine failure (i.e., infants who received the vaccine but failed to mount an effective immune response[54]). Vaccination was assumed to occur when aging from one age group to the next. For the first dose (MCV1), at a given age, individuals were either vaccinated or not vaccinated, determined by the recommended MCV1 age, delay distribution, and MCV1 coverage. If unvaccinated, individuals entered the next age group's susceptible compartment. If vaccinated, the probability of successful vaccination was determined by the VE-age relationship. If successful, infants were protected and entered the recovered compartment of the next age group. If unsuccessful, they remained unprotected and entered the next age group's vaccinated-susceptible compartment. The process remained the same for MCV2, but vaccination occurred when aging from the vaccinated susceptible compartment. The recommended MCV2 age was modeled as 6 months after the recommended MCV1 age, aligning with the modal gap between reported MCV schedules[44–47].

Full model details, including parameterization, are included in the supplementary material (Supplementary Table 2).

### Recapitulating reported pre-vaccine mean ages of measles infection

To calibrate the transmission level of the model, we fitted the vaccine-free MAI to target reported MAI estimates from the pre-vaccine era[23], which ranged between 24 and 72 months (see Supplementary Table 3). We fitted the model to these target MAIs by minimizing the squared difference between the target and modeled MAI, calibrating the parameter $q$ (relative transmissibility of <5-year-olds compared to ≥5-year-olds) for a range of $R_0$ values between 10 to 20. More specifically, we split the pre-vaccine MAI into three groups: high transmission (24–36 months), medium transmission (36–48 months) and low transmission (48–72 months). For each transmission level, we fitted the model to the lower-bound, mid-value, and upper-bound of the MAI range. We found the target MAI could be recapitulated by multiple $R_0$–$q$ pairs. Hence, to capture the range in transmission parameterization, we selected 5 $R_0$–$q$ pairs for each target MAI, resulting in 15 pairs for each transmission level.

The model was run assuming a constant population of 10 million for 500 years, at which point convergence to the equilibrium solution was determined by the magnitude of the derivatives[55]. If this convergence criterion was not fulfilled, the final 20 years of the simulation were extracted, and a linear regression model of time against cases was fit to the modeled cases. The simulation was judged to have converged if the slope of the linear model was less than $10^{-3}$ per day, corresponding to a change of <1 case per year. If convergence was achieved, the modeled MAI was calculated and compared against the target MAI.

As a sensitivity analysis, to test the impact of the choice of 5 years of age as the relative transmissibility cutoff, we repeated the analysis for the South Africa SCM, with $q$ defined as the relative transmissibility of <3-year-olds relative to ≥3-year-olds.

### The optimal age to recommend MCV1

We simulated recommending MCV1 at different ages, monthly from 6 to 20 months. For each recommended age, we calculated the corresponding annual incidence from the final year of the simulation, then identified the MCV1 age that minimized the incidence aggregated over all age groups. Furthermore, to identify factors that have the greatest impact on the optimal age we varied these factors across realistic values (Fig. 1b).

We simulated measles annual incidence using the model described above, simulating without vaccination for 50 years, then introducing MCV1 and MCV2 and running the model for a further 950 years, to achieve equilibrium. Vaccination was modeled as beginning from the recommended MCV1 or MCV2 age, with delays in MCV1 and MCV2 following the delay distributions described above. The vaccine coverage was defined as the proportion of a birth cohort vaccinated by 24 months after the recommended MCV dose age. Based on reported WHO/UNICEF estimates of national MCV immunization coverages[56] in countries with mean annual incidence >1 per 100,000 between 2010 and 2019, we set MCV1 coverage at 45%, 55%, 65%, 75%, and 85%, and MCV2 coverage at 5% points lower than the set MCV1 coverage.

To assess variation in optimal ages, we estimated the optimal age for every combination of SCM (China, India, Japan, Moscow, South Africa, UK, USA), vaccine coverage (MCV1 coverages: 45%, 55%, 65%, 75%, 85%), delay distribution (short-tailed delay and long-tailed delay), VE curve (2.5%, 25%, 75%, 97.5% quantiles, and the MLE), and transmission level (low-, medium-, and high-transmission, with 15 $R_0$–$q$ pairs for each level, see Fig. 1b). Any combination that failed to converge to the equilibrium solution according to the abovementioned convergence criteria, at any recommended MCV1 age, was removed. Optimal ages were then calculated and compared. To facilitate this comparison and evaluate the current WHO clustering of recommendations, for every combination of transmission level and MCV1 coverage we clustered (using PAM clustering[50] based on Euclidean

distance, and the silhouette method[50] to determine cluster sizes) the estimated optimal ages to identify groups of SCMs.

### Assessing time to converge to the minimum incidence

To assess how long it takes for the incidence to converge to the optimal incidence value when changing the MCV1 age, we simulated changing the recommended MCV1 age to the optimal age and then assessed the time for the incidence to converge to within 10% of the optimal incidence. Following the WHO recommendations (87% of reported MCV1 ages), we simulated changing to the optimal age from an initial age of either 9 or 12 months for all parameter sets, excluding sets with the same initial age and optimal age.

### Optimizing the MCV1 ages for social contact matrices from low- and lower-middle-income countries

The SCMs used in the above analysis were selected due to their high age-group resolution, providing contact rates between yearly age groups. However, these age groups were predominantly derived from high- and upper-middle-income countries (as of 2021)[16]. Of the seven SCMs selected, three came from high-income countries (Japan, UK, USA), three came from upper-middle-income countries (China, Moscow (Russia), and South Africa), and only one SCM came from a lower-middle-income country (India). To better examine SCMs from countries with higher measles burdens, i.e., LMICs, we applied our method to four SCMs from countries in sub-Saharan Africa[25], with either low- or lower-middle-incomes, Ghana (lower-middle), Sierra Leone (low), Uganda (low), and Zambia (low). Notably, SCMs from these countries were only available with contact rates aggregated into 5 yearly age groups, leading to greater ambiguity in contacts in very young age groups.

Along with varying the SCM, the population's demographic structure may impact the optimal MCV1 age. To check this, we tested the LMIC SCMs with age demographics corresponding to type 1 (typical of high-income countries, as detailed above) and type 3 (typical of low-income countries, characterized by high infant mortality) survivorship functions[26]. We parameterized the type 3 demography by extracting the yearly age group population sizes between 2010 and 2019 for the 4 LMICs[57], and fitting a generalized additive model (GAM) to the population normalized mean group population size, with logit transformed age group size as a spline of age (98.8% of deviance explained).

We parameterized the model with either type 1 or type 3 demography, assuming a constant population size of 10 million and constant population sizes for each age group. We repeated the above analysis, calibrating the transmission parameters for each SCM for both demographic types. Following Supplementary Table 3, we assumed LMICs had only high transmission. Hence, we limited the target MAIs to the high transmission level (24–36 months), resulting in 15 $R_0$–$q$ transmission pairs for each SCM-demography pair. Finally, we estimated the optimal MCV1 age for every combination of LMIC SCM (Ghana, Sierra Leone, Uganda, Zambia), demographic type (type 1 and type 3), vaccine coverage (MCV1 coverages: 45%, 55%, 65%, 75%, 85%), delay distribution (short-tailed delay and long-tailed delay), VE curve (2.5%, 25%, 75%, 97.5% quantiles, and the MLE), and $R_0$–$q$ pair.

### Numerical implementation

Analysis was carried out using R version 4.1.1[58], using the R package tidyverse[59] version 1.3.1. Partial correlations were calculated using the package ppcor[60] version 1.1. SCAMs were fitted using the package scam[61] version 1.2.12. Lomax distributions were fitted using the algorithm L-BFGS-B[62]. PAM clustering was carried out using the R packages cluster[63] version 2.1.2 and factoextra[64] version 1.0.7. Mantel tests were calculated using the R package vegan[65] version 2.6.2, using Pearson correlation. The measles model was implemented in C and R, using the R package pomp[35] version 3.6. Parameter fitting was carried out using the suplex algorithm[66] in the R package nloptr[67] version 2.0.0. Mixed-effect models were fitted using the package lme4[68] version 1.1.27.1.

Figures were created using the R package ggplot2[69] version 3.5.0, with country boundaries in Fig. 2 taken from the Natural Earth 1:50 m world map, version 2.0.

**Reporting summary**

Further information on research design is available in the Nature Portfolio Reporting Summary linked to this article.

## Data availability

All data used are from published articles and published reports. Sources for measles incidence, recommended MCV1 age, and income group classification are included in Methods: Data on measles incidence, recommended MCV1 age, and income group classification. Details of the MCV vaccine effectiveness data are given in Methods: The relationship between MCV age and vaccine effectiveness. Details of MCV1 delay distribution data are included in Methods: The distribution of MCV1 delay. Sources for the social contact matrices are included in Methods: Recapitulating reported pre-vaccine mean ages of measles infection and Methods: Optimizing the MCV1 ages for social contact matrices from low- and lower-middle-income countries, which also details the sources for the age-group population sizes. Finally, details of MCV coverage sources are included in Methods: The optimal age to recommend MCV1.

## Code availability

Processed data and code are deposited in the Open Research Data Repository of the Max Planck Society, Edmond (https://doi.org/10.17617/3.50X626) with no end date.

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

## Acknowledgements

This work was supported by the Max Planck Society. Computations were performed at the Max Planck Computing and Data Facility (MPCDF). We thank Pejman Rohani for his valuable feedback on the manuscript.

## Author contributions

E.G. and M.d.d.C. conceptualized the project and designed the methods. Model implementation was carried out by E.G., and reviewed by L.A.B.G. E.G. and M.d.d.C. wrote the paper, with support from M.B. and L.A.B.G. M.d.d.C. supervised the project.

## Funding

## Competing interests

The authors declare no competing interests.
