## [Transparent Peer Review file · Nature Communications]

Estimating the optimal age for infant measles vaccination

Corresponding Author: Ms Elizabeth Goult

Version 0:

Reviewer comments:

Reviewer #1

(Remarks to the Author)

Estimating the optimal age for infant measles vaccination

Overall comments

The authors present an approach and set of examples for optimising the recommended age of vaccination for MCV1. This approach considers the risk of infection, parametrised through R_0 , the social contact matrix and existing vaccination coverage, as well as the age-based efficacy of vaccination if given. The article is an interesting union of competing priorities for vaccination and highlights a good point about optimal ages and heterogeneity in existing recommendations. However, the age structure is only partially considered which will have substantial implications for the risk of infection. Finally, the clarity of the ensuing recommendations and how they may be communicated or implemented in practise is not discussed in detail.

Specific comments

Abstract

- On 'regions' be specific on the geography you are referring to
- Line 19: "a key factor..." this sentence has ambiguous subject, suggest cutting in two for clarity
- Line 22: describe type of method eg. statistical, mathematical etc
- Line 23: subject of "this risk" ambiguous, expand to clarify
- Line 25: specify ages of what
- Line 26-27: much of this result comes from the mean age at infection in different transmission settings- it would be good to clarify how your analysis adds to this ie. by emphasising the inclusion of vaccine efficacy by age

Main text

- Line 44: careful discussion is needed of COVID-related disruptions as these were highly variable by region, setting and vaccine
- Line 51: Add further details on vaccine failures for other reasons and the overall prevalence
- Line 64: transmission level is ambiguous and needs to be defined in the introduction and with respect to the method/ model ie. is it R_0 , is it a function of mean age of infection (seems to be this given figures in the SI) or other
- Line 126: Add date ranges of data as well as a list of countries
- Line 126 onwards: a major concern I have is that data is obtained for LMICs on measles, but social contact structures are taken from a few countries including HICs and the population structure is kept uniform (Line 148). This is mixing very different structures and populations and, as R_0 is a function of population structure, will substantially affect transmission. Social contact matrices have been estimated (with limitations) for a number of countries (Prem et al. is one example) and population age structures are available from UNWPP (albeit annual age groups) for WHO member states. As such it should be possible to pair realistic social contact structures with age distributions. This is particularly important for countries where the age distribution is approximately exponential and measles burden is highest- often in LMICs.
- Line 200: Are you simulating to equilibrium? The 950 years seems very particular.
- Line 205: do these coverage values correspond to specific examples eg. from WUENIC?
- Line 314 onwards: This is another concern- in terms of implementing the recommendations or methods from this study, it would be good to include clear thresholds. As the transmission is a function of vaccination and therefore will affect vaccination recommendations, the transmission level itself may not be the best indicator. In recommendations for the frequency of SIAs, WHO uses the MCV1 coverage as a metric [World Health Organization Measles vaccines: WHO position paper.] perhaps you could include some discussion of this/ consider clearer and less circular metrics.
- Line 393: A more complicated model can act as a barrier for action and may not solve some of the communication/

implementation uses that this may present.

- Generally: in terms of examining recommendations, have you considered the situation where countries are grouped by region (perhaps WHO region) and the same minimal optimal vaccination age recommendation is applied? This would simplify the outputs and provide a clearer message on whether existing recommendations are sufficient.

Reviewer #2

(Remarks to the Author)

This study estimated the optimal age for the first dose of measles-containing vaccination (MCV) considering several important factors, including vaccine effectiveness (VE), transmission level, coverage, delay in vaccination, and social contact matrices. The authors applied sufficient analytical techniques and multiple data sources. However, the study design and rationale need further clarification and justifications.

****Major comments:**

- Please clarify the choice of country characteristics targeted in this study. Does it aim to represent a generic country or a specific income group? The delay in receiving MCV1 was based on data from 45 low- and middle-income countries (LMICs), while the social contact matrices were selected mainly from countries with high or upper-middle incomes.

- Introduction, line 73: Why was the Human Development Index (HDI) selected for analysis? What is the rationale or mechanism for how HDI can have an effect on VE?

- Introduction, line 54, and Methods, line 143: How does the duration of maternal antibodies affect the relationship between age and VE, and how is its uncertainty addressed?

- The systematic review by Hughes et al. focused on studies reporting the VE among children older than 9 months old. However, in figure 1, VE among children less than 9 months old were included. Please describe how the data were extracted.

- Parameter q , relative transmissibility of those < 5 years old compared to ≥ 5 years old. How is the cutoff age determined? Considering that children in many countries attend school before 5 years old, a cutoff of < 5 years old might fit better. It would be helpful if the authors could provide potential biological mechanisms for age-related transmissibility, which social contact matrices cannot sufficiently address.

- Methods, line 205: Historically, MCV2 not only had a lower coverage but was also introduced a few years later than MCV1 in many countries. How is the MCV2 introduction being incorporated into the model?

- Since most countries have already introduced MCV1 in their routine immunisation programmes, the information on the potential benefits (e.g. incidence reduction) of shifting to the optimal schedule would be essential in public health policy. The authors could estimate the potential burden averted from the shift in the actual settings.

- In practice, changing the vaccination schedule can be logistically and financially challenging. Children at certain ages may be unprotected because they are not covered in the new or old vaccination schedules, and additional immunisation activities may be needed to fill the immunity gap. Please discuss these policy implications for shifting towards the optimal age for measles vaccination.

- Discussion, lines 340 & 374: One key finding of the study is that the optimal age is associated with the transmission setting, which has been considered in the WHO recommendation. WHO recommends providing measles vaccines to children at 9 and 12 months old in settings with low and high transmission, respectively, while the authors suggest a wider range of recommended ages based on their study findings. The authors should discuss in-depth the pros and cons of the current WHO recommendation compared to the proposed tailoring schedule. Also, the authors suggested frequently reassessing the optimal age for MCV1 vaccination. More discussion is needed on the frequency or timing for conducting the assessment and data requirements.

- Figure 3b: The two MCV1 delay distributions (short and long) not only have different coverage cumulating speeds but also stabilise at different cumulative coverage - how can the two effects be distinguished? Also, the "long delay" curve is added up more quickly than the "short delay" in the first few months after the recommended ages, and the cumulative proportion of vaccination is later reversed. Please explain the potential drivers of these delay patterns and the impacts on the findings.

****Minor comments:**

- Introduction, lines 35-37: The estimated deaths may vary by time. Please report the calendar year of the cited estimates.

- Introduction, line 71: When was the "mean annual incidence" taken for calculating the partial Spearman rank correlation?

- Methods, line 108: Could excluding studies with an age interval of > 3 months introduce bias to the relationship between age at vaccination and VE? For example, studies with a broad age interval might be conducted in resource-limited settings.

- Methods, line 122: Suggest reporting the number of mean MCV2 VE estimates.

- Methods, line 129: Excluding countries with negative median delay in receiving MCV indicates vaccination can only take place after the recommended age. Please state the underlying assumption and the rationale explicitly.
- Methods, line 148: Please elaborate on the "uniform age distribution" and "constant population size". Are they fixed across all age groups and years?
- Methods, lines 173-182: Suggest adjusting the order of sentences and providing more detailed information to improve clarity. First, provide an overview of the calibration targets and methods (including how many stages are involved in the calibration) and name the parameters to be varied. Then, provide details following the order and structure set up in the previous step. Also, what is the purpose of selecting 5 pairs of "R0-q"?
- Methods, line 187: Clarify how 'a linear model fit' is used.
- Methods, lines 193-201: Which year(s) of incidence were minimised to estimate the optimal age for MCV1? Why was such a long period of 950 years needed for model runs following vaccination?
- Methods, line 205: How were these MCV1 coverage levels derived from the WHO data? What countries were covered?
- Methods, line 212: Put "see Figure 1b" in the bracket.
- Discussion, lines 352-353: Clarify how the pre-vaccination levels correlated with the post-vaccination levels.
- Discussion, line 365: Including "age heterogeneities beyond social contacts" could help measles model calibration, but examining how well SCM data capture the contact pattern is also important, especially for children at a young age. Please also comment on the limitations of currently available SCMs.
- Figures 2 & 4(SCM): Different age units are included in this manuscript. Please ensure that the unit is clearly denoted whenever age is mentioned.

Reviewer #3

(Remarks to the Author)

In this paper, the authors use a mechanistic modeling approach to estimate an optimal age range for measles vaccination in infants and toddlers. The goal of this estimation process is to balance the trade-off between later-age vaccination (e.g., lower likelihood of vaccine failure + higher likelihood of infection) and earlier-age vaccination (e.g., higher likelihood of vaccine failure + lower likelihood of infection). The authors find a fairly wide range of optimal ages for measles vaccination (i.e., 6–20 months, as compared to the current recommendation of 12–15 months for first-dose vaccination in the US, for instance), with earlier-age vaccination recommended for communities with higher risk of measles transmission.

Given the persistence of measles worldwide and growing vaccine hesitancy, this is an important research area that warrants further investigation. That said, I have a few considerations that I would like to raise with the authors, which are listed below:

1. The authors assert that vaccine failure contributes to immunity gaps for measles (line 50); however, most MCVs have been shown to be highly efficacious at the population level. Given that vaccine failure is closely correlated with vaccine effectiveness and efficacy, further information is needed to sufficiently bolster this assertion.
2. Relatedly, given that the authors highlight early-age (i.e., arguably, premature) vaccination as a potential cause of vaccine failure, real-world statistics on how frequently premature vaccination occurs would be helpful.
3. The authors note that first-dose age recommendations around the world are fairly homogenous and largely driven by WHO recommendations (line 76). The text at present seems to imply that these recommendations are range-less, and while this is true in some circumstances, it is not in others (with the US recommended range of 12–15 months being one obvious example). The range ultimately proposed by the authors is still significantly wider than existing ranges (such as that in the US), but it would be worth noting that ranges in and of themselves are currently in use in some parts of the world already.
4. While I agree that most measles deaths occur in LMICs (line 127), most vaccinations (per person) occur in high income countries. As a result, within the context of the model, I would like to see the authors incorporate data on measles vaccination delays from high income countries in addition to LMICs. This is particularly important within the setting of high-income countries with considerable spatial heterogeneity in infection risk and vaccination rate, such as the US (where elimination status is currently under threat and endemicity may soon become a reality once again).
5. Though the synthetic populations used to test the authors' modeling approach is interesting and useful (lines 208–212), synthetic representations of real-world populations are a necessary addition from a policy (and application) perspective. Given that the authors are already using real-world contact matrices (line 151) to develop their synthetic populations, I believe the existing modeling structure should allow the authors to simulate disease transmission in synthetic representations of real-world populations via the integration of country-specific transmission risk* and country-specific vaccination data (in addition to country-specific contact matrices). For instance, I would like to see the authors simulate age-based policies (and subsequent impact on disease transmission) in synthetic representations of a variety of real-world countries—perhaps a grid of countries with low, medium, and high risk of measles transmission and/or low, medium, and high rates of measles vaccination—and make policy recommendations on vaccination age ranges for these selected countries given the present state of affairs (with the understanding that these recommendations should be adaptive as infection risk and vaccination rates change). [*It appears that the authors are using country-specific measles incidence in

their model, but incidence alone is not a sufficient measure of transmission risk. At the very least, incidence needs to be adjusted for country-specific population size.]

6. The selection of contact matrices considered gives me pause. The majority of the seven contact matrices selected represent data from high income countries, which is in direct contrast to the delay data that the authors choose to incorporate (i.e., from LMICs exclusively). The explanation provided for selection of these matrices (line 152) is insufficient given the LMIC focus elsewhere in the text.

7. To avoid some of the limitations raised by the authors pertaining to demographic changes over time (line 389) and the impacts that these changes may incur in a 500-year simulation, I would encourage the authors to consider 1000 single-year simulations, especially in the context of the country-specific simulations suggested above.

8. Discussion of the study within the setting of growing vaccine hesitancy is warranted. Do the authors foresee broadening of the recommended age range as improving uptake of the vaccine, potentially by providing a longer time horizon for parents to stagger early childhood vaccines?

Version 1:

Reviewer comments:

Reviewer #1

(Remarks to the Author)

Thank you for your work addressing the comments raised. I do not have further suggestions.

Reviewer #2

(Remarks to the Author)

The authors made reasonable efforts to address the comments and supporting with additional analysis. The rationale of methods and the interpretation of findings have been improved. However, some of my concerns remain:

Line 137 – It is unclear how greater MCV1 delay (> 3 months) is linked with misreported recommendation age. Is there any literature supporting this relationship? Also, could the longer delay of MCV1 affect the modelling of MCV2? e.g. MCV2 is delivered before MCV1.

The authors' response mentioned that SCMs were ultimately fit for POLYMOD. Could you briefly explain the fitting process and compare the differences between SCMs & POLYMOD matrices? This selection of data input can also be included in the discussion section.

Line 434 – The authors performed a nice sensitivity analysis in Supplementary Figure 6, showing that changing the cutoff age for relative transmissibility resulted in different MCV1 optimal ages. Although the qualitative conclusions for the optimal MCV1 age are similar, the uncertainty in the choice of cutoff age and its potential effect need to be highlighted and discussed.

Line 494 – It is more appropriate to describe the assumption for relative transmissibility as an adjustment rather than a correction since the underlying mechanism is unknown and the choice of cutoff age is uncertain.

Figures 3 & 6 – Suggest adding labels a), b), and c) for the subplots to match the descriptions in the legend.

Reviewer #3

(Remarks to the Author)

RE: Reviewer #3, Comment #5

I understand that the research direction proposed above may be out of the scope of what the authors believe is appropriate for this study and would need to be addressed in separate research project. However, if this is the case, it would behoove the authors to make this explicit under a distinct Limitations and/or Future Work subsection in the Discussion section of their manuscript.

In general, a Limitations and/or Future Work subsection would greatly improve the readability of the Discussion. Some of the materials that should appear under these subsections are interspersed throughout the Discussion already, but it would be helpful if they were reported under a designated subsection instead. Furthermore, a good deal of the new analyses conducted (e.g., integration of LMIC contact matrices from Prem et al.)—though extremely valuable in their own right—also introduce their own limitations to the paper that warrant explicit disclosure in a Limitations subsection.

Summary of changes

We thank our three reviewers for their insightful comments. We have undertaken extensive reanalysis to address the concerns raised.

To address the commonly raised issue of a lack of overlap between the contact matrices and the locations with high measles burden, we repeated the analysis using social contact matrices (SCMs) from 4 low- and lower-middle-income countries¹, specifically Ghana, Sierra Leone, Uganda, and Zambia². Furthermore, to better represent populations typical of high measles-burden settings, we also tested the impact of type 3 survivorship demographic structure (approximately exponential distribution) on the optimal age compared to type 1 survivorship³ (the whole population survives to the population's life expectancy, then dies).

For each SCM-demographic type pair, we carried out parameter calibration of the model to pre-vaccine mean ages of infection in high transmission settings (pre-vaccine mean age of infection (MAI): 24-36 months), then simulated incidence for MCV1 ages between 6-20 months. We found our method was successful in estimating an optimal age in each synthetic population, however, the optimal age did not vary substantially between SCMs, potentially due to the high similarity of the matrices (mantel statistic > 0.99 between all SCM pairs). Furthermore, we found that population structure did not significantly affect the optimal age.

These new analyses are now fully detailed in the revised text, with new sections on the methods and results. "Methods: Optimizing the MCV1 ages for social contact matrices from low and lower-middle income countries", "Results: The optimal age is highly similar across lower income social contact matrices", and "Results: Changes to the demographic structure have a substantial impact on the incidence but only a minor impact on the optimal MCV1 ages", along with an additional new Figure (Figure 6), reproduced below for the reviewers' convenience.

Figure 6: Estimating the optimal MCV1 age in high-burden scenarios. a) Estimated annual incidence when recommending MCV1 at ages 6–20 months, with 45% MCV1 coverage. Each line indicates the relative incidence for a given parameter set, relative to the minimum incidence for that parameter set. b) Heatmap of optimal ages. Opacity indicates the proportion of parameter sets with an optimum in a given MCV1 age. For clarity, the results for 55% and 75% vaccination coverage are not displayed. c) Mean annual incidence per 100,000 at the optimal MCV1 age by demographic type for different MCV1 coverage. Points indicate the mean incidence, and lines indicate the 2.5% and 97.5% quantiles.

We have also added further discussion of the policy implications of changing the MCV1 age. Specifically, we discussed the potential for infants to be unprotected due to the changeover period, the potential impacts of schedule changes on vaccine coverage in light of concurrently scheduled vaccines, and vaccine hesitancy. We also considered potential timescales for MCV1 age interventions to take effect and how frequently the

MCV1 age should be reassessed. We provide a point-by-point answer to all the reviewers' comments below.

Reviewer 1

Overall comments

The authors present an approach and set of examples for optimising the recommended age of vaccination for MCV1. This approach considers the risk of infection, parameterized through R_0 , the social contact matrix and existing vaccination coverage, as well as the age-based efficacy of vaccination if given. The article is an interesting union of competing priorities for vaccination and highlights a good point about optimal ages and heterogeneity in existing recommendations. However, the age structure is only partially considered which will have substantial implications for the risk of infection. Finally, the clarity of the ensuing recommendations and how they may be communicated or implemented in practice is not discussed in detail.

We thank the reviewer for their interest in our study. In response to the reviewer's comments and as explained in our general summary of changes above, we conducted extensive new analyses based on social contact matrices and population structures in four low-income countries. We also substantially revised the discussion to clarify the policy implications and logistical aspects of changing the recommended age of vaccination.

Specific comments

Abstract

- On 'regions' be specific on the geography you are referring to

Agreed, we have changed the sentence to:

"The persistence of measles in many countries demonstrates large immunity gaps".

- Line 19: "a key factor..." this sentence has ambiguous subject, suggest cutting in two for clarity

Agreed, we have split the sentence into two sentences:

"MCV impact is determined, in part, by vaccination age. Infants who receive dose 1 (MCV1) at older ages have a reduced risk of vaccine failure, but also an increased risk of contracting infection before vaccination."

- Line 22: describe type of method eg. statistical, mathematical etc

We have updated the abstract to clarify the type of model:

"Here, we designed a new method—based on a mathematical transmission model incorporating realistic vaccination delays and age variations in MCV1 effectiveness".

- Line 23: subject of “this risk” ambiguous, expand to clarify

We expanded the clause to read:

“the MCV1 age risk trade-off”.

- Line 25: specify ages of what

We have specified the ages:

“We predict a large heterogeneity in the optimal MCV1 ages”.

- Line 26-27: much of this result comes from the mean age at infection in different transmission settings- it would be good to clarify how your analysis adds to this ie. by emphasising the inclusion of vaccine efficacy by age.

We agree with this comment. In this sentence, higher transmission referred to either lower vaccination coverage or higher transmission setting, which both resulted in more cases. To clarify this point, we updated this sentence as follows:

“Furthermore, we show that the optimal age depends on the local epidemiology of measles, with a lower optimal age predicted in populations with lower vaccination coverage or higher transmission.”.

Main text

- Line 44: careful discussion is needed of COVID-related disruptions as these were highly variable by region, setting and vaccine

We thank the reviewer for this interesting comment and agree that COVID-related disruptions were variable across regions. We have added specific information to the introduction on disruption to MCV1 coverage and highlighted regional differences.

“These difficulties were compounded during the COVID-19 pandemic, which caused interruptions in routine vaccinations and supplementary immunization activities (SIAs)^{4,5}, with global MCV1 coverage estimated to be 7.9% lower than expected in 2020, and particularly high disruption reported in the Global Burden of Disease super region of South Asia⁵”.

- Line 51: Add further details on vaccine failures for other reasons and the overall prevalence

We have added information on other reasons for vaccine failure, as well as estimates of occurrence:

“Between 2013 and 2017 25% of measles cases were attributed to MCV failure after 2 doses⁶. Several causes may explain these failures (i.e., cold chain storage failure, and host-related factors such as nutrition and immune status^{7,8}), but one avertable cause of these vaccine failures is the vaccination age”.

- Line 64: transmission level is ambiguous and needs to be defined in the introduction and with respect to the method/ model ie. is it R_0 , is it a function of mean age of infection (seems to be this given figures in the SI) or other

Thank you for highlighting this. The transmission level is defined in the methods section according to the mean age of infection and is a function of R_0 and the relative transmissibility of < 5-year-olds, q . We have updated the introduction to introduce this definition earlier:

“Furthermore, location-specific factors, such as transmission level, defined here by the mean age of infection (MAI), are expected to affect this trade-off resulting in different optimal ages^{9,10}.”

- Line 126: Add date ranges of data as well as a list of countries

The countries included in our analysis are shown in Supplementary Figure 3. Date ranges have been added to the main text:

“we obtained data on MCV1 delay between 1996 and 2005 in 45 low and middle-income countries”.

- Line 126 onwards: a major concern I have is that data is obtained for LMICs on measles, but social contact structures are taken from a few countries including HICs and the population structure is kept uniform (Line 148). This is mixing very different structures and populations and, as R_0 is a function of population structure, will substantially affect transmission. Social contact matrices have been estimated (with limitations) for a number of countries (Prem et al. is one example) and population age structures are available from UNWPP (albeit annual age groups) for WHO member states. As such it should be possible to pair realistic social contact structures with age distributions. This is particularly important for countries where the age distribution is approximately exponential and measles burden is highest- often in LMICs.

We thank the reviewer for this comment, which the other reviewers and the editor also raised. Our rationale for using the contact matrices from Mistry et al.’s study was their detailed age resolution (1-year age bands), which, to our knowledge, is the best available in the literature. This fine-grained resolution was critical for our study because we aimed to predict the optimal vaccination in infants. That said, we agree that a shortcoming of Mistry et al.’s study is the absence of data in low-income countries and the lack of data in lower-middle-income countries (with India being the

only country in this group). In response to the reviewer's comment, we have repeated the analysis for SCM from Prem et al. for uniform and approximately exponential population structures. Full details are given above (see our summary of changes above).

- Line 200: Are you simulating equilibrium? The 950 years seems very particular.

Yes, we are simulating equilibrium. 950 years was chosen to give a total simulation time of 1000 years. The text has been updated to clarify this:

"We simulated measles annual incidence using the model described above, simulating without vaccination for 50 years, then introducing MCV1 and MCV2 and running the model for a further 950 years, to achieve equilibrium."

- Line 205: do these coverage values correspond to specific examples eg. from WUENIC?

Good point. Yes, these coverages were based on the WUENIC estimates. This information has been added to the text:

"Based on reported WHO/UNICEF estimates of national MCV immunization coverages¹¹, we set MCV1 coverage at 45%, 55%, 65%, 75%, and 85%,"

- Line 314 onwards: This is another concern- in terms of implementing the recommendations or methods from this study, it would be good to include clear thresholds. As the transmission is a function of vaccination and therefore will affect vaccination recommendations, the transmission level itself may not be the best indicator. In recommendations for the frequency of SIAs, WHO uses the MCV1 coverage as a metric [World Health Organization Measles vaccines: WHO position paper.] perhaps you could include some discussion of this/ consider clearer and less circular metrics.

The reviewer raises an important point. We agree that clear thresholds would be useful to public health authorities. However, our results show that making recommendations based solely on metrics such as incidence would be inappropriate, as vaccination coverage and contact structure substantially impact the optimal age. Furthermore, as our results represent synthetic populations, defining thresholds off them would not give thresholds representative of real-world populations.

Throughout the manuscript, we define transmission levels according to the pre-vaccination MAI, so the transmission is independent of the vaccination coverage. However, we recognize that the pre-vaccination MAI is not always available. Hence, we recommend that the measles model be fitted to time series data on measles cases to accurately characterize the current transmission level in the focal population.

That having been said, model fitting may be considered a barrier to use. Considering the MCV1 coverage and the post-vaccination MAI could allow for the definition of clearer metrics once the model has been applied to multiple real-world scenarios. As shown in Supplementary Figure 11, there is a positive relationship between the post-vaccination MAI and the optimal age. However, this validates the reviewer's intuition and suggests that the post-vaccination MAI (which is a function of vaccination coverage and transmission setting) could be a useful metric to predict the optimal age.

The updated discussion includes further discussion of metrics for the MCV1 age, along with the new Supplementary Figure 11, reproduced below for the reviewer's convenience.

Supplementary Figure 11: Optimal MCV1 age by post-vaccination MAI. The optimal age of a simulated parameter set is shown to increase with post-vaccination MAI across SCM, when controlling for MCV1 age. Data are from simulations at non-optimal MCV1 ages, compared with the corresponding optimal age.

- Line 393: A more complicated model can act as a barrier for action and may not solve some of the communication/ implementation uses that this may present.

We agree with the reviewer that model literacy may be a barrier to action. However, with increasing and cheap access to computing resources, deploying the model we propose in real-world settings is feasible. As a first step in this direction, we have made all the programming codes underlying our mode freely available, with appropriate documentation for users to implement in their settings.

- Generally: in terms of examining recommendations, have you considered the situation where countries are grouped by region (perhaps WHO region) and the same minimal optimal vaccination age recommendation is applied? This would simplify the

outputs and provide a clearer message on whether existing recommendations are sufficient.

The reviewer raises an interesting point. Regarding whether existing recommendations are sufficient, we deliberately did not discuss this point because we only simulated our model in synthetic populations. As discussed in the main text, applying our method to real-world settings will require fitting our model to detailed incidence data to assess the transmission level while considering other complexities (e.g., seasonality, population structure) in a focal population. Hence, while we provide a general method to estimate the optimal age, we cannot evaluate current recommendations.

We agree that grouping countries by region may simplify the model outputs and the recommendations. Such an approach, however, will only be efficient if the grouped countries are similar in every dimension affecting the optimal age, including social contact structure, vaccination coverage, and transmission level. Interestingly, based on our new analyses in low-income countries, we predict comparable optimal ages, suggesting the possibility of common recommendations. This result reflects the homogeneity in their social contact structure, but note that we did not assess potential differences in the other aspects listed above. Future work will be needed to assess this idea. We now elaborate on this point in the discussion:

“Furthermore, as shown in previous analysis¹², SCMs are generally similar across regions; hence it may be possible to make regional recommendations. This would allow countries to assess the MCV1 schedule relatively easily, without ignoring all the potential heterogeneity in optimal ages identified in our analysis.”

Reviewer 2

This study estimated the optimal age for the first dose of measles-containing vaccination (MCV) considering several important factors, including vaccine effectiveness (VE), transmission level, coverage, delay in vaccination, and social contact matrices. The authors applied sufficient analytical techniques and multiple data sources. However, the study design and rationale need further clarification and justifications.

****Major comments:**

- Please clarify the choice of country characteristics targeted in this study. Does it aim to represent a generic country or a specific income group? The delay in receiving MCV1 was based on data from 45 low- and middle-income countries (LMICs), while the social contact matrices were selected mainly from countries with high or upper-middle incomes.

The reviewer raises an important point, which the other reviewers and the editor also highlighted. Our rationale for using the contact matrices from Mistry et al.'s study was their detailed age resolution (1-year age bands), which, to our knowledge, are the best available in the literature. This fine-grained resolution was critical for our study because we aimed to predict the optimal vaccination in infants. That said, we agree that a disadvantage of using the matrices from Mistry et al. is the underrepresentation of low- and middle-income countries, where the majority of measles infections occur. To address the reviewer's comment, we have extended the analysis to apply the method to SCM from low- and lower-middle-income countries. The analysis is detailed more fully above (see our summary of changes above).

Regarding the vaccination delay data, we were constrained by the available data, most of which came from LMICs. However, in response to the reviewer's comment, we identified studies that estimated delays to MCV1 receipt in 3 high-income countries (UK, Norway, and Switzerland). Interestingly, we found no systematic differences in delays between the income groups, except in Switzerland, where the study we identified reported very short delays (see Supplementary Table 1, and Supplementary Figure 4, reproduced below for the reviewer's convenience).

More generally, our study aims to propose a method for calculating the optimal MCV1 age. Applying the method to synthetic populations was intended as a proof of concept and as an exploration of factors that substantially impact the optimal age rather than as a replication of specific countries. As we wrote in the discussion, applying our method to real-world settings will require fitting our model to incidence data to capture the epidemiological dynamics in the focal population.

Supplementary Figure 4: Modeled cumulative fraction vaccinated with MCV1 by month for high-income countries compared to low-income countries. The spread of fitted distributions from low-income countries is indicated by the ribbon. Fitted Lomax distributions from high-income countries are indicated by curves.

- Introduction, line 73: Why was the Human Development Index (HDI) selected for analysis? What is the rationale or mechanism for how HDI can have an effect on VE?

We assume here you meant the effect of the HDI on MCV1 age. HDI was initially selected as a proxy for country income. This was based on the rationale that MCV1 age is better predicted by country income rather than incidence, which would be expected from the WHO recommendations. We have now replaced the HDI with world bank derived income groups, which qualitatively gave the same results:

“The partial Spearman rank correlation coefficient between MCV1 ages and administrative regions’ mean annual incidence between 2010-2019 was only 0.025 (p-value: 0.90) in regions with ≥ 1 measles case per 1 million per year, when controlling for a country’s 2021 World Bank income group classification¹”.

- Introduction, line 54, and Methods, line 143: How does the duration of maternal antibodies affect the relationship between age and VE, and how is its uncertainty addressed?

The reviewer raises an interesting point. In our model, the age–VE relationship is fixed and not directly affected by the duration of maternal antibodies. However, as we discussed in the introduction, the observed increase in VE with age may be interpreted as a consequence of immune system maturation and blunting of the vaccine immune response by maternal antibodies.

The methods have been updated to clarify this:

“The uncertainty in the MCV1 VE relationship may reflect both the variation between infant immune system maturation and in maternal antibody blunting levels. Hence, we can capture the impact of both biological mechanisms by sampling the breadth of uncertainty in the MCV1 VE relationship.”

- The systematic review by Hughes et al. focused on studies reporting the VE among children older than 9 months old. However, in figure 1, VE among children less than 9 months old were included. Please describe how the data were extracted.

Thank you for pointing this out. While it is true that the analysis presented in Hughes et al. 2020 only focused on VE ≥ 9 months, their systematic review identified MCV1 VE estimates for ages ≤ 5 months, with three studies reporting VE estimates for < 9 months. These VE estimates were included in Supplementary data 2 in Hughes et al. 2020¹³. We took the VE estimates directly from this supplementary data.

We have updated the text to clarify how the data were extracted:

“To quantify how vaccination age affects MCV VE, we fitted a statistical model to reported estimates of MCV VE, obtained from a systematic review¹³ (Supplementary data 2.)”

- Parameter q , relative transmissibility of those < 5 years old compared to ≥ 5 years old. How is the cutoff age determined? Considering that children in many countries attend school before 5 years old, a cutoff of < 5 years old might fit better. It would be helpful if the authors could provide potential biological mechanisms for age-related transmissibility, which social contact matrices cannot sufficiently address.

We agree with the reviewer that the choice of the cutoff at 5 years is relatively arbitrary. However, the value was chosen because most children enter school at that age. In

general, diary-based reporting of social contacts may be underreported for young children¹⁴, which may explain the need for the relative transmissibility parameter. (We note that even though Mistry et al.'s method was not diary-based, their contact matrices were ultimately fitted to the POLYMOD matrices, which were diary-based.) Without the inclusion of the parameter q , the model was not flexible enough to recover the target MAI.

As a sensitivity analysis, we repeated the optimization method for the South African SCM for q , defined as the relative transmissibility of < 3 -year-olds compared to ≥ 3 -year-olds. The results are shown in Supplementary Figure 6 (reproduced below, for the reviewer's convenience). They are similar to those of the cutoff at 5 years, though 3 years tended to lead to slightly younger optimal MCV1 ages. However, as the parameter q is slightly arbitrary, this highlights the importance of fitting the model to data when applying the described method to a real-world population.

Reporting of this analysis has been added to the text in "Results: The relative transmissibility parameter cut-off has minor impacts on the optimal age".

Supplementary Figure 6: Heatmap comparing the optimal ages for the South African social contact matrix when the relative transmissibility parameter is defined as ≤ 3 years, versus ≤ 5 years. Opacity indicates the proportion of parameter sets with an optimum at a given MCV1 age. For clarity, the results for 55% and 75% vaccination coverage are not displayed.

-Methods, line 205: Historically, MCV2 not only had a lower coverage but was also introduced a few years later than MCV1 in many countries. How is the MCV2 introduction being incorporated into the model?

This is completely correct. As we were running the simulations to equilibrium, we chose to keep the model as simple as possible and introduce MCV1 and MCV2 simultaneously. However, applications to real-world populations would require that the 2 doses be introduced separately to capture the observed times of MCV1 and MCV2 introduction and the immune landscape of the population. We have clarified the simultaneous introduction in the methods:

“simulating without vaccination for 50 years, then introducing both MCV1 and MCV2 vaccination and running the model for a further 950 years”,

and noted the importance of vaccine introductions in the discussion,

“These include detailed information on past and current MCV1 and MCV2 coverages to build up a detailed picture of the population’s current immune status.”

- Since most countries have already introduced MCV1 in their routine immunisation programmes, the information on the potential benefits (e.g. incidence reduction) of shifting to the optimal schedule would be essential in public health policy. The authors could estimate the potential burden averted from the shift in the actual settings.

The reviewer is exactly right. Table 1 already presents information on the relative incidence increase when recommending at 9 and 12 months versus the optimal age for selected scenarios. In response to the reviewer, we have updated this table to also indicate the absolute difference in incidence (*i.e.*, the number of cases that could be averted). Furthermore, Figure 5a displays the relative annual incidence for each scenario at 45% MCV1 coverage.

- In practice, changing the vaccination schedule can be logistically and financially challenging. Children at certain ages may be unprotected because they are not covered in the new or old vaccination schedules, and additional immunisation activities may be needed to fill the immunity gap. Please discuss these policy implications for shifting towards the optimal age for measles vaccination.

This is a good point that we had not fully addressed. We have added a discussion of the policy implications of changing the MCV1 age to the discussion. We recognize that changing the MCV1 age will incur some expense, but we still expect it to be a relatively low-cost intervention.

- Discussion, lines 340 & 374: One key finding of the study is that the optimal age is associated with the transmission setting, which has been considered in the WHO recommendation. WHO recommends providing measles vaccines to children at 9 and 12 months old in settings with low and high transmission, respectively, while the authors suggest a wider range of recommended ages based on their study findings. The authors should discuss in-depth the pros and cons of the current WHO recommendation compared to the proposed tailoring schedule. Also, the authors suggested frequently reassessing the optimal age for MCV1 vaccination. More discussion is needed on the frequency or timing for conducting the assessment and data requirements.

Thank you for noting this. Because the current results are derived from synthetic populations, we feel that making more direct comparisons would be inappropriate. As we wrote in the discussion, the application of our method in a real-world setting will require fitting our model to incidence data to get a precise picture of the epidemiological dynamics in the target population. However, following sufficient applications of the method to real-world populations, making comparisons would be possible. In this case, as discussed above, it may also be more realistic to make grouped recommendations according to vaccine coverage and post-vaccine mean age of infection. We have added a discussion of this to the text.

To assess the appropriate frequency of reassessment of the MCV1 age, we carried out an additional analysis, quantifying the time to converge to within 10% of the optimal incidence when changing the MCV1 age. We identified approximately 10 yearly to be appropriate, with the caveat that the time to converge is dependent on transmission (with a longer convergence time for lower transmission, as expected from epidemic theory). Full details are given in methods and “Results: The timescale for changes to take effect reduces with increased transmission”, and in Supplementary Figure 14 (reproduced below for the reviewer’s convenience).

Supplementary Figure 14: Time to converge to within 10% of the optimal incidence from 9 or 12 months by SCM and transmission level. Dots indicate median time to converge, lines indicate the interquartile range (IQR).

- Figure 3b: The two MCV1 delay distributions (short and long) not only have different coverage cumulating speeds but also stabilise at different cumulative coverage - how can the two effects be distinguished? Also, the "long delay" curve is added up more quickly than the "short delay" in the first few months after the recommended ages, and the cumulative proportion of vaccination is later reversed. Please explain the potential drives of these delay patterns and the impacts on the findings.

The two delay distributions are captured by the Lomax distribution, a probability distribution, hence the cumulative proportion of the population vaccinated will tend to 100% for large delays. For Figure 3b, we show the unscaled distributions that were fitted to the delay quantiles¹⁵. However, when using the distributions to parameterize the transmission model, we assumed all vaccination occurred within 24 months of the vaccination age, and rescaled the distribution to give the desired vaccination coverage. This is detailed more fully in the supplementary material (Equation 5).

Thank you for pointing out the potential confusion caused by the cluster labels. We have changed the labels to “long-tailed delay” and “short-tailed delay” to clarify the main difference occurring in the right tail (Figure 3, Supplementary Figure 9).

The Lomax distribution is identified by 2 parameters, a shape parameter and a scale parameter, which allow for more flexibility in the function compared to the exponential distribution. Variations in these parameters explain the changes to the shape of the function between clusters, especially the fact that the long-delay distribution is initially above the short-delay one. The social determinants behind the variation are less clear. There are no clear patterns in the distribution clustering of income groups or continents.

Overall, we found only minor impacts of the delay distribution on the optimal age. Of the parameter sets with changes in the delay distribution corresponding to changes in the optimal age (12.2%), the short-tailed delay corresponded to older optimal ages. The short-tailed delay results in the recommended age lying closer to the mean true vaccination age for most of the population, hence meaning the recommended age must be older to result in the optimal true vaccination age.

****Minor comments:**

- Introduction, lines 35-37: The estimated deaths may vary by time. Please report the calendar year of the cited estimates.

Thank you for highlighting this. As it turns out, the initially reported estimate of 2-3 million measles-related deaths per year was misreported in a previous publication, and was in fact the number of deaths prevented between 1999 and 2005 by increased measles immunization efforts. A source for the number of estimated deaths in the early 1960s has been found and reported,

“Historically, measles was a major childhood disease, infecting almost all individuals in early life¹⁶ with over 6 million estimated measles-related deaths occurring per year in the early 1960s¹⁷.”

- Introduction, line 71: When was the "mean annual incidence" taken for calculating the partial Spearman rank correlation?

This is detailed in methods as the mean of the annual measles incidence by country reported to the WHO between 2010 and 2019. We have added the date range to the introduction text to clarify this,

“The partial Spearman rank correlation coefficient between MCV1 ages and countries’ mean annual incidence between 2010-2019 was only 0.025 (p-value: 0.90) in

countries with ≥ 1 measles case per 1 million per year, when controlling for a country's 2021 World Bank income group classification".

- Methods, line 108: Could excluding studies with an age interval of > 3 months introduce bias to the relationship between age at vaccination and VE? For example, studies with a broad age interval might be conducted in resource-limited settings.

This is indeed a potential bias that we didn't fully consider. In response to the reviewer's comment, we reanalyzed the data to check for any patterns in excluded estimates. Out of 105 MCV1 VE estimates from 18 countries, we excluded 36 estimates, from 9 countries due to the age interval ≥ 3 . This criterion completely excluded 4 countries from the analysis: New Zealand (8 estimates, high-income), Singapore (1 estimate, high-income), USA (1 estimate, high-income), and Ukraine (7 estimates, lower-middle-income). The remaining estimates came from 7 high-income countries, 1 upper-middle-income country, 3 lower-middle-income countries, and 3 low-income countries, according to the 2020 World Bank income group classifications¹⁸.

Our complete inclusion criteria resulted in 53 estimates from 16 studies from 12 countries. We have added a new figure (Supplementary Figure 1, reproduced below) detailing the country origins of estimates included and excluded.

Supplementary Figure 1: Vaccine effectiveness estimates¹³ included and excluded from the analysis, according to estimate country. Not shown, 5 excluded estimates from Romania with MCV1 age between 24 and 132 months.

- Methods, line 122: Suggest reporting the number of mean MCV2 VE estimates.

Good point; this has been added to the text:

“For MCV dose 2 (MCV2) VE, too few estimates (17 estimates at 3 unique MCV2 ages with age range < 3 months) were available to assess age dependencies (Supplementary Figure 2).”

-Methods, line 129: Excluding countries with negative median delay in receiving MCV indicates vaccination can only take place after the recommended age. Please state the underlying assumption and the rationale explicitly.

The countries excluded from the analysis were Colombia and Dominican Republic. Both countries reported a target age of 12 months and median delays of -3.8 months

and -4.4 months, respectively, potentially indicating a misreporting of the target age. We have updated the text to better reflect our rationale here,

“ To avoid potentially misreported recommended MCV1 ages, we excluded any countries with an absolute median delay > 3 months (see Supplementary Figure 3).”

- Methods, line 148: Please elaborate on the "uniform age distribution" and "constant population size". Are they fixed across all age groups and years?

Yes, exactly. The total population size remains constant with time, and so does each age group's population. Similarly, we parameterized the model to keep the age cohort size constant until death at 80 years of age. The text has been updated to clarify this:

“We assumed a type 1 survivorship³ (i.e., all individuals achieve the population life expectancy, then die), corresponding to equally sized uniform age distribution 5-yearly age groups, and constant overall population size”

Please note that in the revised submission (see our summary of changes above), we conducted new simulations with type 3 (i.e., exponentially decreasing with age) population structures.

- Methods, lines 173-182: Suggest adjusting the order of sentences and providing more detailed information to improve clarity. First, provide an overview of the calibration targets and methods (including how many stages are involved in the calibration) and name the parameters to be varied. Then, provide details following the order and structure set up in the previous step. Also, what is the purpose of selecting 5 pairs of "R0-q"?

We have reordered the paragraph as advised by the reviewer:

“To calibrate the transmission level of the model, we fitted the vaccine-free MAI to target reported MAI estimates from the pre-vaccine era¹⁹, which ranged between 24 and 72 months (see Supplementary Table 3). We fitted the model to these target MAIs by minimizing the squared difference between the target and modeled MAI, calibrating the parameter q (relative transmissibility of <5 year-olds compared to ≥ 5 year-olds) for a range of R_0 values between 10–20. More specifically, we split the pre-vaccine MAI into three groups: high transmission (24–36 months), medium transmission (36–48 months) and low transmission (48–72 months). For each transmission level, we fitted the model to the lower-bound, mid-value, and upper-bound of the MAI range. We found

the target MAI could be recapitulated by multiple R_0 - q pairs. Hence, to capture the range in transmission parameterization, we selected 5 R_0 - q pairs for each target MAI, resulting in 15 pairs for each transmission level."

- Methods, line 187: Clarify how 'a linear model fit' is used.

We clarified this point as follows:

"the final 20 years of the simulation were extracted, and a linear regression model of time against cases was fit to the modeled cases".

- Methods, lines 193-201: Which year(s) of incidence were minimised to estimate the optimal age for MCV1? Why was such a long period of 950 years needed for model runs following vaccination?

The final year of the simulation was used to calculate the annual incidence. A period of 950 years post-vaccination was chosen to result in an overall simulation time of 1000 years. This total simulation time was chosen to allow the model to converge from as many parameter sets as possible. The text has been changed to clarify the incidence optimized over:

"we calculated the corresponding annual incidence from the final year of the simulation".

- Methods, line 205: How were these MCV1 coverage levels derived from the WHO data? What countries were covered?

MCV1 coverage levels were derived to cover the full range of MCV1 coverages which resulted in measles incidence > 1 per 100,000, without resulting in elimination. The text has been updated to clarify this:

"Based on reported WHO/UNICEF estimates of national MCV immunization coverages in countries with mean annual incidence > 1 per 100,000 between 2010 – 2019, we set MCV1 coverage at 45%, 55%, 65%, 75%, and 85%, and MCV2 coverage at 5% points lower than the set MCV1 coverage."

- Methods, line 212: Put "see Figure 1b" in the bracket.

Changed as suggested.

- Discussion, lines 352-353: Clarify how the pre-vaccination levels correlated with the post-vaccination levels.

We clarified this point as follows:

“This explains the effect of pre-vaccination transmission levels (controlled by the parameters R_0 and q), which are correlated with post-vaccination transmission levels (Spearman rank correlation between pre- and post-vaccination MAI: 0.76).”

- Discussion, line 365: Including "age heterogeneities beyond social contacts" could help measles model calibration, but examining how well SCM data capture the contact pattern is also important, especially for children at a young age. Please also comment on the limitations of currently available SCMs.

This is an important point, as diary-based surveillance methods often underreport contacts of young children¹⁴. As all contact matrices used in the analysis were based, in part, on contact diaries (because all the contact matrices from Mistry et al. were ultimately fitted to the diary-based POLYMOD matrices), this is a potential bias in these SCMs. We have added comments on this to the discussion:

“This heterogeneity could also be interpreted as a correction to the SCM, either because the SCMs used were derived for more modern populations than the pre-vaccine MAI estimates, or due to systematic biases resulting from diary-based reporting of social contacts¹⁴”.

- Figures 2 & 4(SCM): Different age units are included in this manuscript. Please ensure that the unit is clearly denoted whenever age is mentioned.

Good point, we have updated Figures 2 and 4 accordingly.

Reviewer 3

In this paper, the authors use a mechanistic modeling approach to estimate an optimal age range for measles vaccination in infants and toddlers. The goal of this estimation process is to balance the trade-off between later-age vaccination (e.g., lower likelihood of vaccine failure + higher likelihood of infection) and earlier-age vaccination (e.g., highlighter likelihood of vaccine failure + lower likelihood of infection). The authors find a fairly wide range of optimal ages for measles vaccination (i.e., 6–20 months, as compared to the current recommendation of 12–15 months for first-dose vaccination in the US, for instance), with earlier-age vaccination recommended for communities with higher risk of measles transmission.

Given the persistence of measles worldwide and growing vaccine hesitancy, this is an important research area that warrants further investigation. That said, I have a few considerations that I would like to raise with the authors, which are listed below:

1. The authors assert that vaccine failure contributes to immunity gaps for measles (line 50); however, most MCVs have been shown to be highly efficacious at the population level. Given that vaccine failure is closely correlated with vaccine effectiveness and efficacy, further information is needed to sufficiently bolster this assertion.

We agree with the reviewer that MCVs are, overall, highly efficacious. However, this efficacy is also known to vary with age⁷, resulting in vaccine failures. Furthermore, these vaccine failures are estimated to contribute substantially to reported measles cases⁶. Even in highly vaccinated populations, failure to seroconvert can be sufficient to reduce the protected prevalence to levels low enough to lead to measles outbreaks^{20,21}.

We have updated the introduction to clarify the impacts vaccine failures have on measles control:

“Although the immunity gaps that drive continued measles transmission are mainly caused by insufficient vaccine coverage, they also result from vaccine failures. Between 2013 and 2017 25% of measles cases were attributed to MCV failure after 2 doses⁶. Several causes may explain these failures (i.e, cold chain storage failure, and host-related factors such as nutrition and immune status^{7,8}), but one key avertable cause of these vaccine failures is the vaccination age”.

2. Relatedly, given that the authors highlight early-age (i.e., arguably, premature) vaccination as a potential cause of vaccine failure, real-world statistics on how frequently premature vaccination occurs would be helpful.

As shown in Figure 2, two countries report to the WHO recommending MCV1 age younger than 9 months: South Africa at 6 months and China at 8 months¹¹. Hence, these two countries may be considered as examples of early-age vaccination, when vaccine effectiveness is not yet maximal (see Figure 3). To clarify this point, statistics on vaccine failure have been added to the introduction:

“Between 2013 and 2017, 25% of measles cases were attributed to MCV failure after 2 doses⁶.”

3. The authors note that first-dose age recommendations around the world are fairly homogenous and largely driven by WHO recommendations (line 76). The text at present seems to imply that these recommendations are range-less, and while this is true in some circumstances, it is not in others (with the US recommended range of 12–15 months being one obvious example). The range ultimately proposed by the authors is still significantly wider than existing ranges (such as that in the US), but it would be worth noting that ranges in and of themselves are currently in use in some parts of the world already.

Thank you for pointing this out. To account for ranges in the recommended MCV1 age, we repeated the MCV age recommendation extraction from the WHO and ECDC databases and identified 16/210 administrative regions that recommend an age in MCV1 ages (India, Republic of Korea, Turkmenistan, Austria, Cyprus, Czech Republic, Finland, Greece, Italy, Latvia, Lithuania, Poland, Slovakia, Slovenia, USA, and Canada), of which 9 recommended a range of 3 months, including the USA.

We have detailed how we dealt with ranges in the text:

“In cases where a range of MCV1 ages were recommended we extracted the minimum age.”,

and updated Figure 2 to capture any changes from the re-extraction.

4. While I agree that most measles deaths occur in LMICs (line 127), most vaccinations (per person) occur in high income countries. As a result, within the context of the model, I would like to see the authors incorporate data on measles vaccination delays from high income countries in addition to LMICs. This is particularly important within the setting of high-income countries with considerable spatial heterogeneity in infection risk and vaccination rate, such as the US (where elimination status is currently under threat and endemicity may soon become a reality once again).

This is an important limitation of the data on delay distribution, which has also been raised by the other reviewers. To address this important comment, we identified delay distributions from high-income countries (HICs). Unexpectedly, the information on vaccination delay was more readily available for LMICs than HICs, and we could only

identify estimates from 3 countries (see Supplementary Table 1). We checked the delay distributions by fitting Lomax distributions to the HIC data, and, comparing the distributions, found the distributions were of a similar scale to LMICs, except for Switzerland which reported very short delays (see Supplementary Figure 4, reproduced below for the reviewer’s convenience).

Supplementary Figure 4: Modeled cumulative fraction vaccinated with MCV1 by month for high income countries compared to low income countries. The spread of fitted distributions from low income countries is indicated by the ribbon. Fitted Lomax distributions from high income countries are indicated by curves.

5. Though the synthetic populations used to test the authors’ modeling approach is interesting and useful (lines 208–212), synthetic representations of real-world populations are a necessary addition from a policy (and application) perspective. Given that the authors are already using real-world contact matrices (line 151) to develop their synthetic populations, I believe the existing modeling structure should allow the authors to simulate disease transmission in synthetic representations of real-world populations via the integration of country-specific transmission risk* and country-specific vaccination data (in addition to country-specific contact matrices). For instance, I would like to see the authors simulate age-based policies (and subsequent impact on disease transmission) in synthetic representations of a variety of real-world countries—perhaps a grid of countries with low, medium, and high risk of measles transmission and/or low, medium, and high rates of measles vaccination—and make policy recommendations on vaccination age ranges for these selected countries given

the present state of affairs (with the understanding that these recommendations should be adaptive as infection risk and vaccination rates change). [*It appears that the authors are using country-specific measles incidence in their model, but incidence alone is not a sufficient measure of transmission risk. At the very least, incidence needs to be adjusted for country-specific population size.]

We generally agree that our approach of simulating synthetic populations should ultimately be expanded to simulate actual populations and make recommendations in real-world settings. As we wrote in the discussion, however, the application of our method in a real-world setting will require extending our model (to capture, for example, seasonality and stochasticity) and fitting it to age-specific incidence data (which, in many locations, are not directly available) to get a precise picture of the epidemiological dynamics in the target population. In our experience, such fitting is extremely long and computationally intensive (see Domenech de Cellès et al. 2018²² for an overview of the effort required in a *single* location) and would represent a separate research project. In addition, we believe such an exercise is beyond the scope of our study, which mainly aims to detail a new model and method to estimate the optimal age.

That said, in the revised manuscript, we present new simulations based on other demographic structures (with type 3 or exponential mortality). We hope these new analyses will help at least partly address the reviewer's comment.

6. The selection of contact matrices considered gives me pause. The majority of the seven contact matrices selected represent data from high income countries, which is in direct contrast to the delay data that the authors choose to incorporate (i.e., from LMICs exclusively). The explanation provided for selection of these matrices (line 152) is insufficient given the LMIC focus elsewhere in the text.

This is an important point that was raised by the editor and the other reviewers. The previous social contact matrices were chosen because of the high age group resolution, i.e., yearly age groups. Given our general goal to estimate the optimal MCV1 age in infants, this fine-grained age resolution was critical for our study. We have conducted an additional analysis using contact matrices from low and lower-middle-income countries, taken from Prem et al. (2021). The disadvantage of using these contact matrices is the lower age group resolution (5-yearly age groups), which partly conceals the nuances of the age group contacts, particularly in younger age groups. More information on the analysis is detailed above (see our summary of changes).

7. To avoid some of the limitations raised by the authors pertaining to demographic changes over time (line 389) and the impacts that these changes may incur in a 500-year simulation, I would encourage the authors to consider 1000 single-year

simulations, especially in the context of the country-specific simulations suggested above.

In our simulations, we do not model demographic changes, but instead keep the population and demography constant. We have clarified this in the methods:

“We assumed type 1 survivorship³ (i.e., all individuals achieve the population life expectancy, then die), corresponding to equally sized 5-yearly age groups, and constant overall population size”.

We also repeated the analysis, changing the demographic age structure to type 3 survivorship, but also kept the overall population size and age-specific population sizes constant throughout the simulations.

Simulating for 500 and 1,000 years was chosen to ensure convergence of the simulations to the equilibrium incidence. 500 years was used when calibrating the model transmission, to reduce simulation times. In this case, all the fitted simulations fulfilled our convergence criteria.

On the other hand, we used 1000 years when simulating vaccinating the population because the models took longer to converge to equilibrium (see below).

Total daily cases simulated using the Moscow SCM at 85% MCV1 coverage in a low transmission setting, with long-tailed delay distribution for a single transmission parameter set.

8. Discussion of the study within the setting of growing vaccine hesitancy is warranted. Do the authors foresee broadening of the recommended age range as improving uptake of the vaccine, potentially by providing a longer time horizon for parents to stagger early childhood vaccines?

This is an important point we indeed neglected to consider. Older or broader schedule recommendations may lead to increased MCV1 uptake, as a subset of vaccine-hesitant parents choose to vaccinate their children on a delayed “alternative” schedule²³. We have added a discussion of this and the potential trade-off between optimal age and increased MCV1 coverage to the discussion:

“In light of this, the relationship between vaccine hesitancy and vaccine schedules should be carefully considered. A subset of vaccine-hesitant parents choose not to completely forgo vaccination, but rather to delay vaccines. If older MCV1 ages lead to increased coverage, it may be more advantageous to recommend a non-optimal MCV1 age to maximize coverage.”

References

1. World Bank. World Bank income groups – with major processing by Our World in Data. *Our World in Data* <https://ourworldindata.org/grapher/world-bank-income-groups?tab=table&time=2021..latest> (2023).
2. Prem, K. *et al.* Projecting contact matrices in 177 geographical regions: An update and comparison with empirical data for the COVID-19 era. *PLOS Comput. Biol.* **17**, e1009098 (2021).
3. Demetrius, L. Adaptive value, entropy and survivorship curves. *Nature* **275**, 213–214 (1978).
4. World Health Organization. Measles and rubella strategic framework 2021-2030. *Measles and rubella strategic framework: 2021-2030* <https://www.who.int/publications-detail-redirect/measles-and-rubella-strategic-framework-2021-2030> (2020).
5. Causey, K. *et al.* Estimating global and regional disruptions to routine childhood vaccine coverage during the COVID-19 pandemic in 2020: a modelling study. *The Lancet* **398**, 522–534 (2021).
6. Patel, M. K. & Orenstein, W. A. Classification of global measles cases in 2013–17 as due to policy or vaccination failure: a retrospective review of global surveillance data. *Lancet Glob. Health* **7**, e313–e320 (2019).
7. Moss, W. Measles in Vaccinated Individuals and the Future of Measles Elimination. *Clin. Infect. Dis.* **67**, 1320–1321 (2018).
8. Fappani, C. *et al.* Breakthrough Infections: A Challenge towards Measles Elimination? *Microorganisms* **10**, 1567 (2022).
9. McLean, A. R. & Anderson, R. M. Measles in developing countries Part I. Epidemiological parameters and patterns. *Epidemiol. Infect.* **100**, 111–133 (1988).
10. Mclean, A. R. & Anderson, R. M. Measles in developing countries. Part II. The predicted impact of mass vaccination. *Epidemiol. Infect.* **100**, 419–442 (1988).
11. World Health Organisation. WHO Immunization Data portal. [https://immunizationdata.who.int/compare.html?COMPARISON=type1__WIISE/MT_AD_COV_LONG+type2__WIISE/MT_AD_INC_RATE_LONG+option1__MCV_coverage+option2__MEASLES_incidence&GROUP=Countries&YEAR=.](https://immunizationdata.who.int/compare.html?COMPARISON=type1__WIISE/MT_AD_COV_LONG+type2__WIISE/MT_AD_INC_RATE_LONG+option1__MCV_coverage+option2__MEASLES_incidence&GROUP=Countries&YEAR=)
12. Mistry, D. *et al.* Inferring high-resolution human mixing patterns for disease modeling. *Nat. Commun.* **12**, 323 (2021).
13. Hughes, S. L. *et al.* The effect of time since measles vaccination and age at first dose on measles vaccine effectiveness – A systematic review. *Vaccine* **38**, 460–

- 469 (2020).
14. Mossong, J. *et al.* Social Contacts and Mixing Patterns Relevant to the Spread of Infectious Diseases. *PLOS Med.* **5**, e74 (2008).
 15. Clark, A. & Sanderson, C. Timing of children's vaccinations in 45 low-income and middle-income countries: an analysis of survey data. *The Lancet* **373**, 1543–1549 (2009).
 16. Langmuir, A. D. Medical Importance of Measles. *Am. J. Dis. Child.* **103**, 224–226 (1962).
 17. Wolfson, L. J. *et al.* Has the 2005 measles mortality reduction goal been achieved? A natural history modelling study. *The Lancet* **369**, 191–200 (2007).
 18. World Bank Country and Lending Groups – World Bank Data Help Desk. <https://datahelpdesk.worldbank.org/knowledgebase/articles/906519-world-bank-country-and-lending-groups>.
 19. Anderson, R. & May, R. Vaccination against rubella and measles: Quantitative investigations of different policies. *J. Hyg. (Lond.)* **90**, 259–325 (1983).
 20. Hahné, S. J. M. *et al.* Measles Outbreak Among Previously Immunized Healthcare Workers, the Netherlands, 2014. *J. Infect. Dis.* **214**, 1980–1986 (2016).
 21. Bassal, R. *et al.* Seropositivity of measles antibodies in the Israeli population prior to the nationwide 2018 – 2019 outbreak. *Hum. Vaccines Immunother.* **17**, 1353–1357 (2021).
 22. Domenech de Cellès, M., Magpantay, F. M. G., King, A. A. & Rohani, P. The impact of past vaccination coverage and immunity on pertussis resurgence. *Sci. Transl. Med.* **10**, eaaj1748 (2018).
 23. Gowda, C. & Dempsey, A. F. The rise (and fall?) of parental vaccine hesitancy. *Hum. Vaccines Immunother.* **9**, 1755–1762 (2013).

Reviewers' comments

Reviewer 1:

Reviewer #1 (Remarks to the Author):

Thank you for your work addressing the comments raised. I do not have further suggestions.

Reviewer #1 (Remarks on code availability):

I have checked the code is accessible and legible but have not reimplemented it. A clear readme is included and the scripts are organised in a logical way.

We thank the reviewer for their final comments.

Reviewer 2:

The authors made reasonable efforts to address the comments and supporting with additional analysis. The rationale of methods and the interpretation of findings have been improved. However, some of my concerns remain:

We thank the reviewer for acknowledging our efforts to address their comments. Please find below our replies to their remaining comments.

Line 137 – It is unclear how greater MCV1 delay (> 3 months) is linked with misreported recommendation age. Is there any literature supporting this relationship? Also, could the longer delay of MCV1 affect the modelling of MCV2? e.g. MCV2 is delivered before MCV1.

For the countries excluded (Colombia and the Dominican Republic), the median MCV1 delay was < -3 months. Both countries suffered from measles outbreaks just before the DHS surveys and implemented additional immunization activities to counteract this¹. Because these vaccinations were not carried out following the routine vaccine schedule, the recommended age has close to no impact on the age at vaccination in these circumstances. Hence, we chose to exclude these countries from the analysis.

We have updated the methods to clarify this (Lines: 160–163):

“We excluded two countries (Colombia and the Dominican Republic) from the analysis, as both countries’ delay distributions were affected by changes to the vaccination programs in response to local measles outbreaks”.

Thanks to the implementation of MCV2 vaccination in the model, it is not possible for MCV2 to be received before MCV1. Because we only model MCV2 vaccination in MCV1 failures, they must, by definition, already have received MCV1.

The authors’ response mentioned that SCMs were ultimately fit for POLYMOD. Could you briefly explain the fitting process and compare the differences between SCMs & POLYMOD matrices? This selection of data input can also be included in the discussion section.

In the study from Mistry et al.², the matrices were generated from synthetic populations, specifically generating synthetic households, schools, workplaces, and others, from micro and macro population data. From these synthetic locations, Mistry et al. calculated the location-specific contact matrices. Overall contact matrices were then generated using a weighted linear combination of the location-specific matrices, with the weights determined by multiple linear regression fitting the overall SCM to empirically derived contact matrices from POLYMOD.

In response to the reviewer, we added a paragraph to the discussion concerning this point (Lines: 639–675):

“Even for fixed transmission levels and vaccination coverages, the impact of different social contact structures was strong. This impact demonstrates that, in addition to quantifying the transmission of measles, detailed knowledge of social contact structure, quantified by data-driven SCMs, is needed to identify the optimal age in a given population. In our analysis, we used three sources of SCM. The first³ generated SCM from diary-based reporting for limited locations, while the second² and third⁴ SCM sources either fit to or extrapolated from the diary-based SCM to infer SCM for more locations, hence inheriting the limitations inherent to diary-based SCM alongside method-specific issues. The second SCM source² resulted in highly resolved age-specific contact rates but only applied the method in a low number of LMICs. The third source⁴ had better coverage of LMICs but a lower age resolution, potentially masking differences between countries’ social contact structures and optimal ages. Overall, this indicates the need for highly resolved data-driven SCM, particularly for regions with the greatest burden of infectious diseases, namely LMICs, to more accurately identify the optimal MCV1 ages.”

Line 434 – The authors performed a nice sensitivity analysis in Supplementary Figure 6, showing that changing the cutoff age for relative transmissibility resulted in different MCV1 optimal ages. Although the qualitative conclusions for the optimal MCV1 age are similar, the uncertainty in the choice of cutoff age and its potential effect need to be highlighted and discussed.

We agree with this comment, and to better highlight this point, we revised the results (Lines: 553–556):

“However, variation in the cut-off age led to slight variation in the optimal ages, highlighting the importance of tailoring our measles transmission model based on detailed measles data in the study population to accurately determine the optimal age.”

and the discussion (Lines: 633–636):

“However, as demonstrated by our simulations using an alternative relative transmissibility cutoff (3 years old) successfully reproducing the target MAIs, other definitions of this parameter are possible, and the choice should be based on age-specific factors affecting measles in the study population.”

Line 494 – It is more appropriate to describe the assumption for relative transmissibility as an adjustment rather than a correction since the underlying mechanism is unknown and the choice of cutoff age is uncertain.

Thank you for pointing this out. We have changed the description to “adjustment”.

Figures 3 & 6 – Suggest adding labels a), b), and c) for the subplots to match the descriptions in the legend.

Thank you for catching this. We have added the labels.

Reviewer 3:

Reviewer #3 (Remarks to the Author):

RE: Reviewer #3, Comment #5

I understand that the research direction proposed above may be out of the scope of what the authors believe is appropriate for this study and would need to be addressed in separate research project. However, if this is the case, it would behoove the authors to make this explicit under a distinct Limitations and/or Future Work subsection in the Discussion section of their manuscript.

In general, a Limitations and/or Future Work subsection would greatly improve the readability of the Discussion. Some of the materials that should appear under these subsections are interspersed throughout the Discussion already, but it would be helpful if they were reported under a designated subsection instead. Furthermore, a good deal of the new analyses conducted (e.g., integration of LMIC contact matrices from Prem et al.)—though extremely valuable in their own right—also introduce their own limitations to the paper that warrant explicit disclosure in a Limitations subsection.

We thank the reviewer for this helpful comment. In response, we substantially revised the discussion to structure it into sub-sections (Lines: 587–887), including one about the limitations (entitled “Limitations and further requirements for applying the method to real-world settings”, Lines: 686–792). We also included the suggested further analysis in the discussion (Lines: 739–744):

“While making recommendations based on our current synthetic simulations would be inappropriate, future work based on our method may allow for more generalizable recommendations. For example, systematic application of our method to near-real-world populations for various SCM, vaccine coverages, and measles risk levels would illuminate the range of realistic and observable optimal ages.”

References:

1. Clark, A. & Sanderson, C. Timing of children's vaccinations in 45 low-income and middle-income countries: an analysis of survey data. *The Lancet* **373**, 1543–1549 (2009).
2. Mistry, D. *et al.* Inferring high-resolution human mixing patterns for disease modeling. *Nat. Commun.* **12**, 323 (2021).
3. Mossong, J. *et al.* Social Contacts and Mixing Patterns Relevant to the Spread of Infectious Diseases. *PLOS Med.* **5**, e74 (2008).
4. Prem, K. *et al.* Projecting contact matrices in 177 geographical regions: An update and comparison with empirical data for the COVID-19 era. *PLOS Comput. Biol.* **17**, e1009098 (2021).